# Neutrophil extracellular traps promote metastasis in gastric cancer patients with postoperative abdominal infectious complications

Xiang Xia[1,4], Zizhen Zhang[1,4], Chunchao Zhu[1,4], Bo Ni[1], Shuchang Wang[1], Shuofei Yang[2], Fengrong Yu[1], Enhao Zhao[1✉], Qing Li[3✉] & Gang Zhao [1✉]

Postoperative abdominal infectious complication (AIC) is associated with metastasis in locally advanced gastric cancer (GC) patients after radical gastrectomy. However, the underlying mechanism remains unclear. Herein, we report that neutrophil extracellular traps (NETs), the DNA meshes released by neutrophils in response to infection, could promote GC cells proliferation, invasion, migration and epithelial–mesenchymal transition dependent on TGF-β signaling. Then we model nude mice with cecal puncture without ligation to simulate postoperative AIC and find that NETs in peripheral blood and ascites fluid facilitate GC cells extravasation and implantation into liver and peritoneum for proliferation and metastasis. Notably, TGF-β signaling inhibitor LY 2157299 could effectively impede liver and peritoneal metastasis but not concurrently aggravate sepsis in those AIC-bearing nude mice. These findings implicate that targeting downstream effectors of NETs such as TGF-β signaling might provide potential therapeutic prospect to reduce the risk of GC metastasis.

---

[1] Department of Gastrointestinal Surgery, Ren Ji Hospital, School of Medicine, Shanghai Jiao Tong University, Shanghai, People's Republic of China. [2] Department of Vascular Surgery, Ren Ji Hospital, School of Medicine, Shanghai Jiao Tong University, Shanghai, People's Republic of China. [3] State Key Laboratory of Oncogenes and Related Genes, Shanghai Cancer Institute, Ren Ji Hospital, School of Medicine, Shanghai Jiao Tong University, 200240 Shanghai, People's Republic of China. [4]These authors contributed equally: Xiang Xia, Zizhen Zhang, Chunchao Zhu. ✉email: zhaoenhao@renji.com; 1385160147B@163.com; zhaogangrj@163.com

   1

Gastric cancer (GC) is the fifth most common malignancy and the fourth leading cause of cancer-related mortality worldwide in 2018[1]. Radical gastrectomy + D2 lymphadenectomy still remain the mainstay of curatively therapeutic management for locally advanced GC[2]. However, even though undergoing R0 resection (no cancer residue after resection microscopically), 20–60% locally advanced GC patients would suffer from recurrence or metastasis[3,4]. Generally, GC recurrence or metastasis are largely attributed to tumor depth, lymph node status, cancer biology, aggressiveness and the radicality of surgery[5]. However, recent studies, including our own research[6], revealed that postoperative complications (Clavien-Dindo ≥ II)[7–9], especially infectious complications[10] such as anastomotic leak[11] or pneumonia[12] had also been reported to promote tumor recurrence or metastasis and thus decrease long-term survival in locally advanced GC patients. Understanding how postoperative infectious complications facilitate recurrence or metastasis is imperative to develop therapeutic interventions.

Neutrophils, the most abundant leukocytes in peripheral blood, are considered as the first immune cells to function against infectious or inflammatory insults[13,14]. These polymorphonuclear leukocytes are well known for their antimicrobial activity via phagocytosis, degranulation of cytotoxic enzymes and proteases, and neutrophil extracellular traps (NETs)—scaffolds of chromatin with associated cytotoxic enzymes and proteases that are released into the extracellular space where they can trap microorganisms[15,16]. Recently, a growing body of evidence indicates a role for NETs in promoting cancer metastasis, including establish metastatic foci[17,18], awakening dormant cancer cells[19] or inducing cancer cell chemotaxis[20]. Moreover the strategies, such as neutrophils depletion or NETs digestion were reported to powerfully abate the accelerated metastasis in previously mentioned studies. However, those NETs inhibition strategies to prevent metastasis were hardly applicable to surgical patients, since either neutrophils or NETs play principal roles in postoperative wound healing, inflammatory responses and postoperative sepsis during perioperative course[21].

In this study, we show that abdominal infectious complication (AIC) after gastrectomy would stimulate neutrophils to release NETs both in peripheral blood and abdominal cavity. Furthermore, those AIC-induced NETs could facilitate GC metastasis in vitro and in vivo dependent on TGF-β signaling. Notably, we find that TGF-β inhibitor LY 2157299 is a potential therapeutic to decrease metastasis without exacerbating sepsis in the setting of liver and peritoneal metastasis nude mice with AIC-induced NETs, which raise the possibility to prolong the survival of GC patients.

## Result

**Postoperative AIC reduces overall survival (OS) and recurrence-free survival (RFS) in locally advanced GC patients undergoing R0 resection.** Firstly, we retrospectively analyzed clinicopathological characteristics and prognosis data from 1315 pathological stage (pStage) I-III GC patients undergoing R0 resection in Renji hospital from January 2010 to December 2015. In this GC cohort, a total of 215 cases experienced postoperative complications and as shown in Supplementary Fig. 1A, their 5-year OS rate and RFS were significantly lower than those without postoperative complications (OS: 47.0% vs. 61.4%, $p = 0.001$; RFS: 38.1% vs. 58.8%, $p = 0.001$). Moreover, their corresponding survival curves classified by pStage (I, II, III) were presented graphically in Supplementary Fig. 1B. Notably, the significantly shorter 5-year OS and RFS due to complication only existed in II and III cohort. Further stratified analysis indicated that those significant adverse effects on prognosis mainly resulted

from postoperative AIC rather than Non-AIC (Supplementary Fig. 1C, D). Those above results pointed that postoperative AIC was a risk variable for decreased OS and RFS in locally advanced GC patients.

**Postoperative AIC stimulates NETs release both in peripheral blood and abdominal cavity.** To determine whether postoperative AIC could stimulate neutrophils to release NETs, we isolated neutrophils from peripheral blood of control, Non-AIC and AIC groups respectively and then settled those neutrophils on coverlips for 2 h for NETs observation via immunofluorescently co-staining DNA, Cit-H3 (citrullinated histone-3) and MPO (myeloperoxidase). The purity of neutrophils by isolation method were demonstrated more than 90% (Supplementary Fig. 2A, B). Figure 1A showed that compared to neutrophils from control or Non-AIC groups, neutrophils from AIC group released a great number of NETs, which could be specifically digested by DNase I. Furthermore, both serum and plasma NETs levels—which were evaluated by detecting the MPO–DNA complex—were significantly higher in GC patients experiencing AIC compared with those without AIC (control and Non-AIC groups). Consistent results were also observed in ascites fluid (Fig. 1C, D) and omental tissues (Fig. 1E, F), both of which were collected preoperatively and postoperatively by operation or drainage tube (Supplementary Fig. 2C). Moreover, IL-8, a potent neutrophil chemotactic factor, was significantly elevated in ascites fluid collected from AIC group (Supplementary Fig. 2D), that would recruit more neutrophils to abdominal infectious sites for anti-infection (Supplementary Fig. 2E and Fig. 1C–F). These findings indicated that postoperative AIC could stimulate neutrophils to release NETs not only in peripheral blood but also in abdominal cavity.

**NETs-formation neutrophils promote GC cells proliferation, invasion, and migration in vitro and metastasis in vivo.** We next focused on whether AIC-induced NETs-formation neutrophils contributed to GC cells proliferation or metastasis. After coculturing MKN-45 and MGC-803 cells with indicated neutrophils or control, we noted that neutrophils from AIC group promoted GC cells proliferation, invasion and migration compared to those from control and Non-AIC group (Supplementary Fig. 3A and Fig. 2A, B). However, those above enhanced effects could be significantly abolished when PAD4 inhibitor (PAD4i), NE inhibitor (NEi) or DNase I were added to specifically block NETs (Supplementary Fig. 3A and Fig. 2A, B). Of note, neutrophils supernatant from AIC group had no effect on proliferation, invasion or migration of GC cells (Supplementary Fig. 3B–D). Following scan electron microscopy (SEM) and enzyme-linked immunosorbent assay (ELISA) for NETs detection further confirmed that NETs induced pro-invasion and promigration probably depended on NETs presence because little NETs or MPO–DNA were detected in neutrophils supernatant even from AIC group (Supplementary Fig. 3E, F). Moreover, simply coculturing with GC cells could not stimulate neutrophils to release NETs (Supplementary Fig. 3G), which indicated that those above functional NETs were mostly induced by infection rather than GC cells. In order to confirm above findings in vivo, we introduced liver metastasis (LM) and peritoneal metastasis (PM) nude mice models in our study. As a result, coculturing with NETs-formation neutrophils from AIC groups promoted GC cells metastasis, and the treatments that specifically inhibiting NETs could significantly alleviate those metastasis (Fig. 2C, D). Collectively, those data suggested that NETs-formation neutrophils derived from infectious subjects promoted GC cells

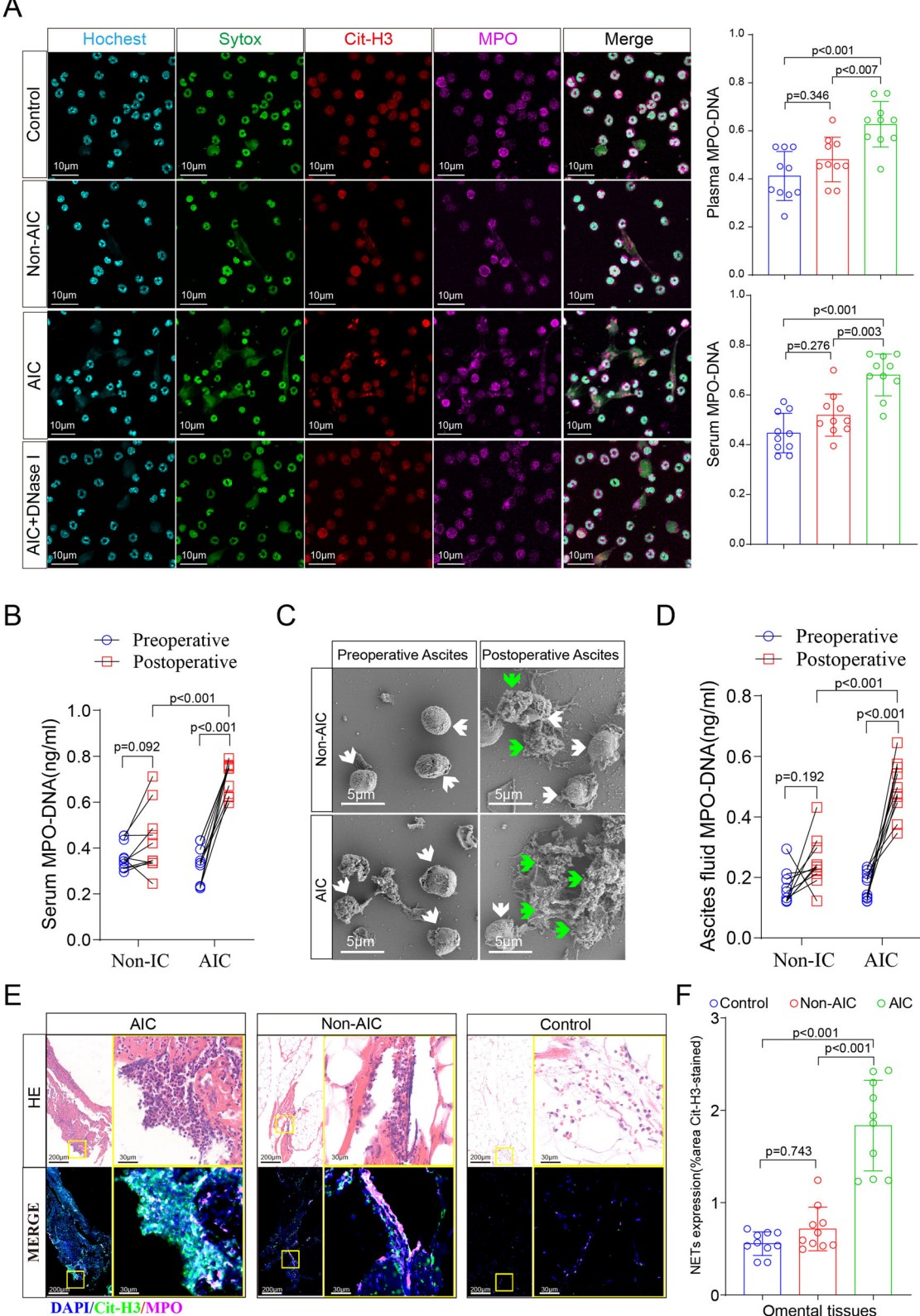

proliferation, migration and invasion in vitro and metastasis in vivo.

**NETs-formation neutrophils modulate EMT in GC cells.** Given that acquisition of epithelial–mesenchymal transition (EMT) phenotype is a critical process for cancer cells to metastasis, we

then examined whether NETs modulated EMT in GC cells. As shown in Fig. 3A, B, MKN-45 and MGC-803 cells that cocultured with neutrophils from AIC group expressed a lower level of epithelial marker E-cadherin and higher level of mesenchymal marker N-cadherin (protein and mRNA) compared to those coculturing with control or Non-AIC group. In addition, specifically inhibiting NETs by PAD4i, NEi or DNase I reversed the

**Fig. 1 Postoperative AIC stimulates NETs release both in peripheral blood and abdominal infectious site. A** Left panel: representative immunofluorescence co-staining of DNA (Hochest and Sytox), Cit-H3 (citrullinated histone-3) and MPO (myeloperoxidase) to assess NETs formation in the neutrophils isolated from peripheral blood of control, Non-AIC, AIC and AIC + DNase I groups; Right panel: plasma and serum levels of MPO–DNA in GC patients with control, Non-AIC and AIC; **B** Preoperative and postoperative serum MPO–DNA levels between Non-AIC and AIC groups; **C** Representative SEM (scan electron microscopy) of neutrophils isolated from preoperative and postoperative ascites fluid between Non-AIC and AIC groups. Green arrows point to extracellular meshes of NETs and white arrows point to neutrophils; **D** Preoperative and postoperative ascites fluid MPO–DNA levels between Non-AIC and AIC groups; **E** Representative images of HE and immunofluorescence staining of DNA, Cit-H3, and MPO in omental tissues among AIC, Non-AIC, and control groups; **F** Quantification of NETs in omental tissues among AIC, Non-AIC, and control groups. Data represent the mean ± S.D. in **A**, **F** ($n = 10$ per group); one-way ANOVA with Tukey test was used in **A**, **F**; paired and unpaired Student's $t$-tests were used in **B**, **D** ($n = 10$ per group). Source data are provided as a Source Data file.

effects of NETs on EMT phenotype (Fig. 3A, B). In accordance with in vitro results, reduced E-cadherin and increased N-cadherin were observed in liver and peritoneum metastatic lesions collected from LM and PM nude mice (Fig. 3C). To further corroborate above experimental findings clinically, we collected liver and peritoneum metastatic lesions from GC patients. As shown in Fig. 3D, metastatic lesions with higher NETs signals exhibited lower E-cadherin and higher N-cadherin, and vice versa. Moreover, GC patients with more NETs in metastatic lesions parallelly exhibited higher NETs in the plasma and serum (Fig. 3D). Collectively, NETs-formation neutrophils promoted metastasis was associated with EMT program of GC cells.

**NETs induced proliferation, invasion, migration, and EMT of GC cells depend on TGF-β signaling pathway activation.** TGF-β signaling pathway is the most important activator for EMT, which has also been reported to correlate with several cancers progression and metastasis[22], including GC[23,24]. In consideration of TGF-β1 is the most prevalent isoform and consistently and significantly correlated with TGF-β signaling activation in many human tumor types (TCGA gene expression data)[25], we used TGF-β1(TGF-β agonist) and LY 2157299 (TGF-β inhibitor) to identify the roles of TGF-β signaling in NETs mediated EMT as well as invasion, migration, and proliferation (Fig. 4 and Supplementary Fig. 4). As a result, MKN-45 and MGC-803 cells showed increased expression of N-cadherin, p-Smad2 (nuclear) and reduced expression of E-cadherin upon TGF-β1 stimulation in cell immunofluorescence assays, and those effects could be reversed by TGF-β inhibitor LY 2157299 (Supplementary Fig. 4A, B). Likewise, NETs mediated E-cadherin downregulation, N-cadherin and p-Smad2 (nuclear) upregulation were abrogated after LY 2157299 treatment (Fig. 4A, B). Western blot assays further confirmed above results that the alterations of E-cadherin, N-cadherin, total p-Smad2/3 and nuclear p-Smad2/3 due to NETs presence could be reversed by LY 2157299 treatment (Fig. 4A, B). In transwell and proliferation assays, both NETs and TGF-β1 enhanced invasion, migration and proliferation were significantly abated by LY 2157299 treatment (Fig. 4C, D and Supplementary Fig. 4C–E). To determine whether TGF-β1 level was positively correlated with NETs formation, we tested TGF-β1 level in serum and ascites fluid from indicated GC patients. Figure 4E showed that the serum as well as ascites fluid TGF-β1 level from AIC group was significantly higher than those from control and Non-AIC groups. Then correlation analysis revealed that the serum and ascites fluid TGF-β1 level showed good agreement with corresponding MPO–DNA levels (Fig. 4F). However, TGF-β1 level did not elevate in neutrophils supernatant from AIC group or neutrophils medium that cocultured with GC cells, whose results were consistent with their NETs level (Supplementary Fig. 3F, G). Immunofluorescence co-staining DNA, Cit-H3 and TGF-β1 in neutrophils (Fig. 4G) and clinical samples (Fig. 4H) further demonstrated that TGF-β1 expression was positively synchronous with NETs release.

Together, these data implicated that NETs induced proliferation, invasion, migration and EMT of GC cells were dependent on TGF-β signaling activation.

**NETs-GC clusters facilitate GC cells extravasation and implantation at metastatic sites.** NETs have been reported to promote tumor progression via binding lung and ovarian cancer cells[18,26]. Here, we designed a specific transwell experiment (Fig. 5A) to investigate whether NETs binding to GC cells could also account for the metastasis-supporting ability of neutrophils from AIC group. As shown in Figure 5B, C, D, significantly more penetrated GC cells were found in the membranes when a mixture of MKN-45 cells and NETs-formation neutrophils added to the upper chamber compared to those without NETs-formation neutrophils mixture (Fig. 5B) or treatment with TGF-β inhibitor (Fig. 5D). Notably, Fig. 5C exhibited the representative images of NETs trapping GC cells via examining and observing lower chamber medium by SEM: significantly more GC cells were attached to the webs of NETs to develop into NETs-GC clusters in the presence of NETs-formation neutrophils while little NETs-GC cluster formed in those without NETs-formation neutrophils.

Cecal ligation and puncture (CLP) is the most widely used mouse model to study the pathogenesis of infection[27], which could also greatly simulates postoperative AIC of patients. In our study, a modified infection model that nude mice underwent 25G needle cecal puncture without ligation (CP) was designed to study the effects of postoperative AIC-induced NETs on GC cells in vivo (Supplementary Fig. 5A). Supplementary Fig. 5B, C demonstrated that all ten CP (25G) mice survived 30 days with a sepsis status while only 50% (21G) and 80% (25G) CLP mice were alive at post-modeling day 30. PET/CT images (Supplementary Fig. 5D) as well as serum and ascites fluid MPO–DNA levels (Supplementary Fig. 5E) further corroborated that CP nude mice suffered AIC-induced NETs release both in abdominal cavity and peripheral blood. In order to investigate the role of NETs-GC clusters in vivo, we injected GFP-MKN-45 cells into spleen (LM nude mice model) or abdominal cavity (PM nude mice model) of sham, CP (25G) and CP (25G) + DNase I nude mice. Then at post-modeling day 1 and day 5, we collected liver, peritoneum, serum and ascites fluid samples for GC cells or NETs examination. As a result, significantly more GC cells accompanied with higher NETs expression were observed in the liver and peritoneum of CP (25G) groups as compared to sham or CP (25 G) + DNase I groups (Fig. 5E, F and Supplement Fig. 6A, B). Additionally, the NETs and TGF-β1 level in serum as well as ascites fluid from CP (25G) groups were also significantly elevated (Supplementary Fig. 6C, D). Moreover, neutrophils isolated from CP (25G) nude mice could also form NETs-GC clusters in the specific transwell assay as that from GC patients did (Supplementary Fig. 6E, F). Altogether, these above suggested implicated that AIC-induced NETs could trap free GC cells to form NETs-GC clusters and then contributed to GC cells extravasation and implantation in those metastatic organs.

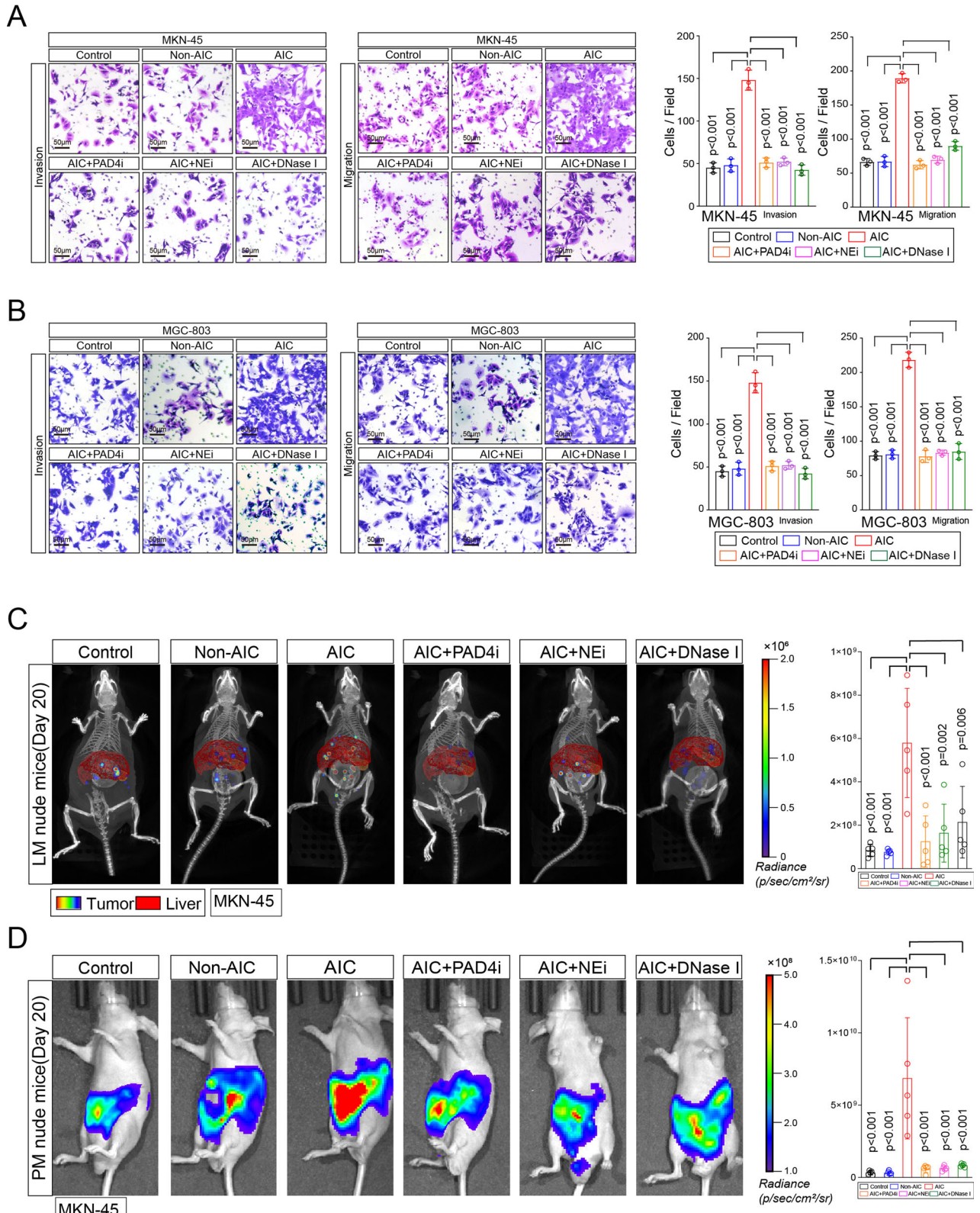

**Fig. 2 NETs promote GC cells invasion and migration in vitro and metastasis in vivo. A**, **B** MKN-45 (**A**) and MGC-803 (**B**) cells as indicated neutrophils and treatment were subjected to transwell Matrigel invasion and migration assays; **C**, **D** Representative images (left) and quantification (right) of metastatic lesions in luciferase-MKN-45 cells bearing LM (**C**) and PM (**D**) nude mice models. After coculturing with neutrophils from indicated groups and treatment, luciferase-MKN-45 cells were injected to spleen (LM model) or abdominal cavity (PM model) for metastasis. Data represent the mean ± S.D. in **A**, **B** (n = 3 biologically independent experiments) and **C**, **D** (n = 5 per group); one-way ANOVA with Tukey test was used in **A**, **B**, **C**, **D**. Source data are provided as a Source Data file.

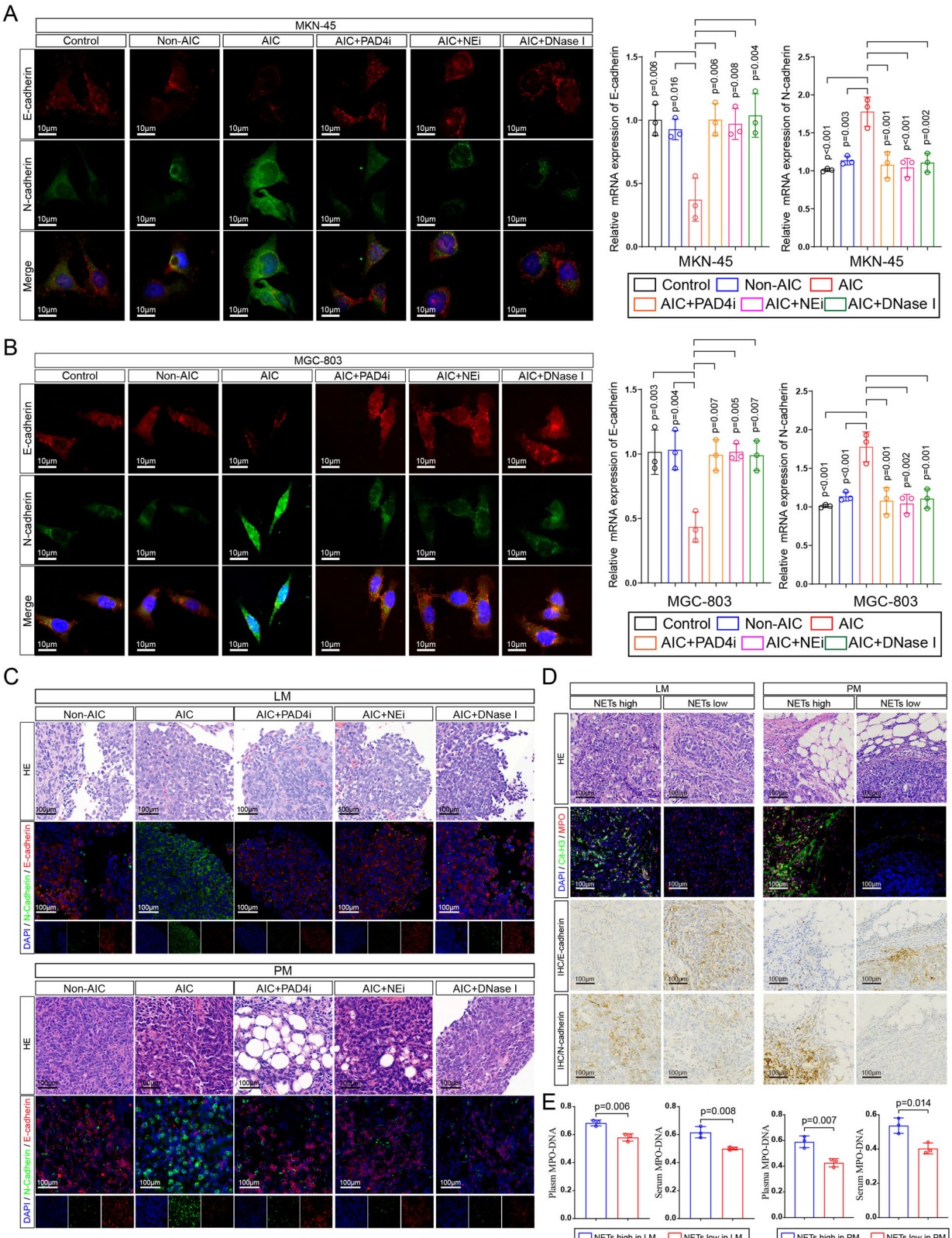

**Fig. 3 NETs modulate EMT in GC cells. A**, **B** Left panel: representative immunofluorescence co-staining of DNA, E-cadherin and N-cadherin in MKN-45 (**A**) and MGC-803 (**B**) cells as indicated treatments (PADi, NEi, or DNase I were used to inhibit NETs specifically). Right panel: The mRNA levels of E-cadherin and N-cadherin in MKN-45 and MGC-803 cells as indicated treatments were assayed by quantitative reverse transcriptase (qRT)–PCR; **C** Representative HE (metastatic lesions) and immunofluorescence co-staining images of DNA, E-cadherin and N-cadherin in liver (upper panel), and peritoneum (down panel) collected from LM and PM nude mice ($n = 5$ per group) as indicated in Fig. 2C, D; **D** Representative HE, immunofluorescence co-staining images of DNA, Cit-H3, and MPO, immunohistochemical images for E-cadherin and N-cadherin in metastatic leasions from GC patients with LM ($n = 3$) or PM ($n = 3$); **E** Plasma and serum levels of MPO–DNA in GC patients as indicated in Fig. 3D. Data represent the mean ± S.D. in **A**, **B** ($n = 3$ biologically independent experiments) and **D**, **E** ($n = 3$ per group); one-way ANOVA with Tukey test was used in **A**, **B**; Unpaired Student's *t*-tests were used in **E**. Source data are provided as a Source Data file.

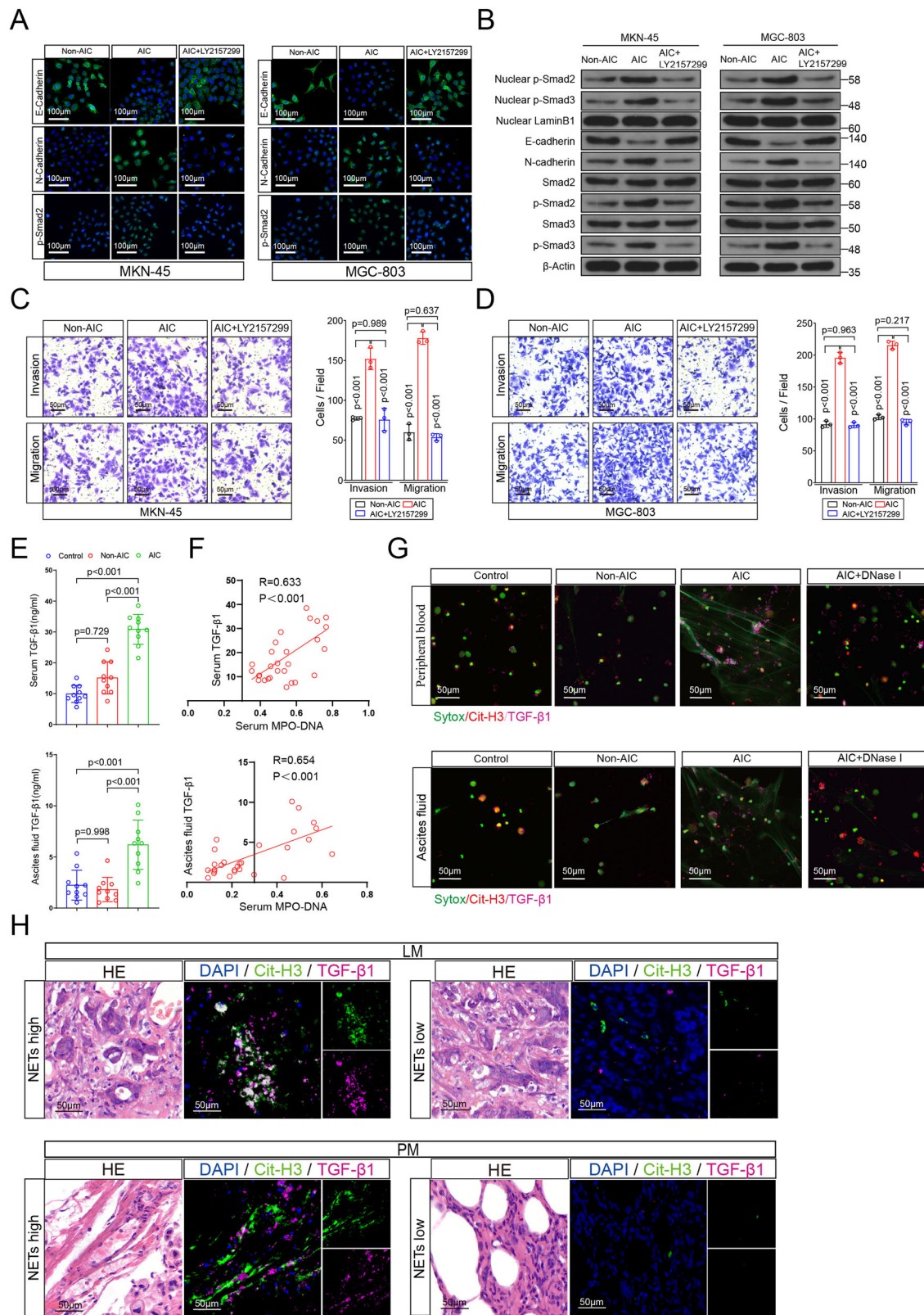

**TGF-β inhibitor effectively counteracts NETs-GC clusters induced metastasis but not aggravated sepsis**. To further explore NETs-GC clusters involved GC metastasis and its potential therapeutics in vivo, an experiment scheme was designed including sham or CP and LM or PM modeling and then treatment, including PBS, DNase I, or LY 2157299 were administered intraperitoneally every 5 days (Fig. 7A). Figure 7B

and Supplementary Fig. 7A, B showed that DNase I administration to inhibiting NETs resulted in overwhelming sepsis and death for CP nude mice during post-modeling 20 days. Yet, nude mice receiving CP or CP + LY 2157299 experienced similar death rate and clinical scores during post-modeling 20 days. For GC metastasis, DNase I or LY 2157299 administration significantly reduced metastatic burden both in LM (Fig. 7C, D) and PM

**Fig. 4 NETs induced invasion, migration and EMT of GC cells dependent on TGF-β signaling pathway activation. A**, **B** Left panel: representative immunofluorescence staining was used to compare E-cadherin, N-cadherin and p-smad2 in MKN-45 (**A**) and MGC-803 (**B**) cells as indicated treatment (LY 2157299, a TGF-β inhibitor). Right panel: Western blotting was used to compare E-cadherin, N-cadherin, Smad3, p-Smad3, nuclear p-Smad3, Smad2, p-Smad2, and nuclear p-Smad2 in MKN-45 (**A**) and MGC-803 (**B**) cells as indicated treatment; **C**, **D** Transwell Matrigel invasion and migration assays for MKN-45 (**C**) and MGC-803 (**D**) cells as indicated treatment. **E** Serum and ascites fluid TGF-β1 levels in GC patients with control, Non-AIC and AIC groups; **F** Correlation between serum MPO–DNA and serum TGF-β1, ascites fluid MPO–DNA and ascites fluid TGF-β1 levels in GC patients with control ($n = 10$), Non-AIC ($n = 10$) and AIC ($n = 10$), the Pearson's correlation coefficient R-value and the p-value are shown in the figures; **G** Representative immunofluorescence co-staining images of DNA, Cit-H3, and TGF-β1 to located NETs and TGF-β1 in the neutrophils isolated from peripheral blood (upper panel) and ascites fluid (down panel) of control, Non-AIC, AIC and AIC + DNase I groups; **H** Representative HE and immunofluorescence co-staining images of DNA, Cit-H3 and TGF-β1 in LM and PM from GC patients. Data represent the mean ± S.D. in **C**, **D** ($n = 3$ biologically independent experiments) and **E** ($n = 10$ per group); one-way ANOVA with Tukey test was used in **C**, **D**, **E**. **A**, **B**, **C**, **D**, **G**, **H** were representative of three biologically independent experiments. Pearson correlation coefficient analysis were used in **F**. Source data are provided as a Source Data file.

(Fig. 7G, H) nude mice models. More importantly, LY 2157299 administration did not interfere NETs formation in CP nude mice, since NETs and TGF-β1 levels in metastatic lesions (Fig. 7E and I, and Supplementary Fig. 7C, D), peripheral blood (Fig. 7F) and ascites fluid (Fig. 7J) of CP + LY 2157299 group were comparable to those of CP group. In addition, ki67 immunohistochemistry (IHC) assay displayed that NETs promoted GC cells proliferation in metastatic sites and this effect would be abolished when NETs digested by DNase I or TGF-β signaling inhibited by LY 2157299 (Supplementary Fig. 7E). Altogether, these data suggested that TGF-β inhibitor effectively counteracts NETs-GC clusters induced metastasis but not aggravated sepsis.

## Discussion

In past decades, the traditional standpoint that neutrophils are merely a bystander in cancers has been revolutionized, and recognition that neutrophils make significant contributions in the initiation, development and progression of cancers has been established[28]. Initially, a higher neutrophils-to-lymphocytes ratio (NLR) or infiltrating tumor-associated neutrophils (TANs) were considered as a strong predictor for poor prognosis[29,30]. Then following researches suggested that TANs were able to present two different polarization states, which were N1(anti-tumor) or N2(pro-tumor)[31,32]. Nevertheless, similar to M1 versus M2 macrophage classification in malignancy[33], the theory concerning TANs dichotomy is probably also an oversimplification since N1 and N2 only represent the laboratory extremes of a biological continuum[34]. Therefore, defining neutrophils upon their distinct functional phenotypes or subpopulations that aid or abate the process of tumourigenesis (e.g., proliferation, invasion, migration and metastatic seeding) in terms of specific features in tumor microenvironment (e.g., primary tumor, circulation, pre-metastatic and metastatic) may be more appropriate[35].

Recently, a growing number of evidences demonstrated that NETs played a significant role not just in infections or non-infectious inflammatory diseases[15,36,37], but also in tumor progression and metastasis[18–20,26,38,39]. Nevertheless, the mechanism underlying AIC-induced NETs mediating GC metastasis are still not clear. By analyzing clinical data and prognosis from GC cohort undergoing R0 resection, we corroborated postoperative AIC such as gastrointestinal leak or abdominal abscess or infection were significantly associated with early recurrence or metastasis of locally advanced GC patients. More importantly, we found postoperative AIC could stimulate neutrophils to release NETs both in peripheral blood and ascites fluid, and then those NETs would trap free GC cells to form NETs-GC clusters in which NETs facilitated GC cells metastasis (Fig. 7).

Consistent with previous researches[18–20,26,38,39], our findings supported NETs promoted cancer metastasis. Moreover, this effect could be diminished by NETs inhibition or neutrophils depletion. However, in clinical practice, especially in postoperative

management, we can hardly eliminate neutrophils or NETs because they are the predominant defense line of innate immune response to microbial pathogens[21]. In accordance with Meng et al.'s study[40], our in vivo experiments also demonstrated NETs inhibition by DNase I administration in the nude mice with AIC predictably aggravated sepsis and sequentially increased death rate. These results implicated that simply inhibiting NETs to alleviate metastasis were not applicable to postoperative patients. Hence, exploiting a potential therapy such as target downstream effectors of NETs on cancer cells should have profound significance for cancer patients who would suffer surgery related infection.

In our study, enhanced proliferation, invasion, migration, mesenchymal phenotype as well as TGF-β signaling activation were observed in GC cells coculturing with NETs-formation neutrophils, yet neither with NETs supernatant nor NETs inhibition. Then the discoveries that TGF-β1 was positively associated with NETs presence and NETs trapped GC cells further consolidated that NETs promoted GC cells metastasis probably dependent on direct contact between NETs and free GC cells, namely developing NETs-GC clusters. Afterward, TGF-β signaling, a widely acknowledged metastasis contributor in GC[23,24], were significantly activated in those GC cells and TGF-β signaling inhibitor LY 2157299 could effectively abate the metastasis. Intriguingly, Szczerba et al. found neutrophils could directly contact circulating tumor cells (CTCs) of breast cancer and thereafter form CTC-neutrophil clusters to expand their metastatic potential[41]. However, whether those CTC-associated neutrophils released NETs or not were not examined. In the light of previous studies that coculturing breast cancer cells with neutrophils could stimulate neutrophils to release NETs[18,39], It was reasonable to deduce that those CTC-associated neutrophils would release NETs to trap breast cancer cells for pro-metastasis, same as NETs-GC clusters proposed in our study (Fig. 7). However, in our study, simply coculturing GC cells with control neutrophils failed to stimulate neutrophils to release NETs, which subsequently come to the conclusion that those pro-metastatic NETs was primarily regulated by infection other than cancer cells.

In some previous studies, NETs in the supernatant ("free NETs") have been shown to bind to cancer cells[42–44] and even promoted breast cancer cells invasion in vitro[39]. While in those study, the isolated "free NETs or NETs-supernatant" were artificially generated by calcium ionophore[43,44], LPS, PMA[39,42], or cancer cells[39] stimulation in vitro, which largely differed from our methods by which NETs were observed or examined without any artificial stimulation except pathophysiological events such as AIC in patients or CP in nude mice. Moreover, both SEM and ELISA (MPO–DNA level) results confirmed that little "free NETs" were detected in the supernatant of neutrophils from AIC or coculturing with GC cells in our study. Despite those above discrepancies in "free NETs", our results, along with other studies, supported that "tangible" NETs could bind to cancer cells and then promote metastasis.

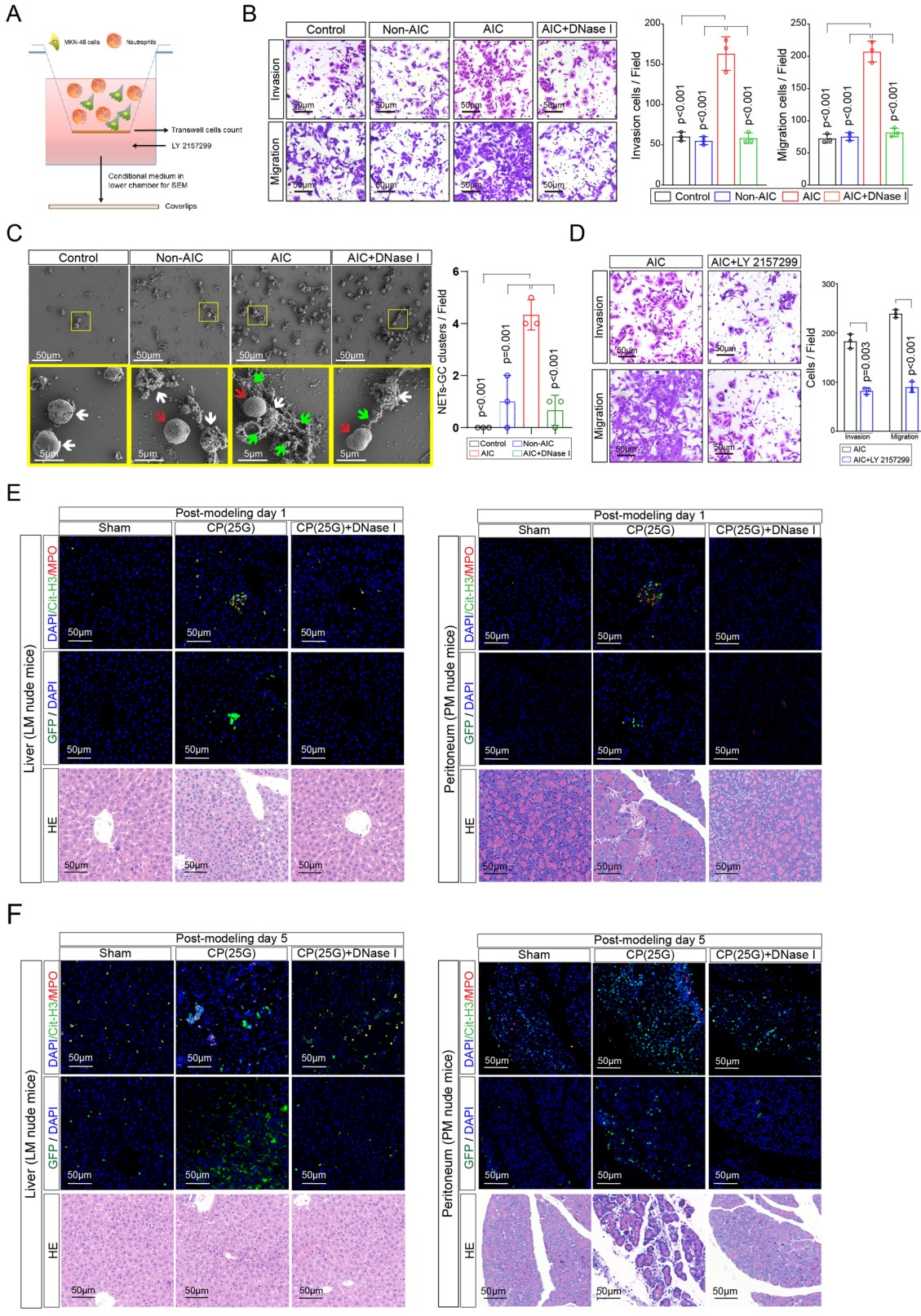

As surgery remains the only potentially curable treatment for most solid tumors, infectious complications or diseases were still the inevitable events in most postoperative management. Our present results emphasized the clinical importance of preventing post-operative AIC, especially in locally advanced GC patients and shed light on the mechanism that postoperative AIC rendered metastasis.

Since depleting neutrophils or inhibiting NETs are not clinically feasible due to risk of infection or sepsis, our findings that TGF-β signaling may be targeted therapeutically to reduce metastasis in the context of metastasis-supporting NETs-GC clusters provides the prospect to mitigate the adverse oncologic consequences in locally advanced GC patients experiencing postoperative AIC.

**Fig. 5 NETs-GC clusters facilitate GC cells extravasation and survival at metastatic sites. A–D** Schematic experiment design for NETs-GC clusters in a specific transwell assay (**A**). The mixture of MKN-45 cells and neutrophils from indicated groups were subjected to upper chamber. Then the penetrated NETs-GC clusters in upper chamber membrane were counted under a microscopy (**B**) and that in the lower chamber medium were analyzed by a SEM, in which green arrows point to extracellular meshes of NETs, white arrows point to MKN-45 cells and red arrows point to neutrophils (**C**); **D** The mixture of MKN-45 cells and neutrophils from AIC group were subjected to transwell assays with or without LY 2157299 added to the lower chamber medium. Then penetrated cells in the upper chamber membrane were counted under a microscopy; **E** Representative immunofluorescence co-staining images of DNA, Cit-H3, and MPO to assess NETs and GC cells extravasation in the liver of LM nude mice (left panel) or implantation in the peritoneum of PM nude mice (right panel) at post-modeling day 1 ($n = 3$ per group); **F** Representative immunofluorescence co-staining images of DNA, Cit-H3 and MPO to assess NETs and GC cells extravasation in the liver of LM nude mice (left panel) or implantation in the peritoneum of PM nude mice (right panel) at post-modeling day 5 ($n = 3$ per group). Data represent the mean ± S.D. in **B**, **C**, **D** ($n = 3$ biologically independent experiments), one-way ANOVA with Tukey test was used in **B**, **C**. paired Student's $t$-tests were used in **D**. Source data are provided as a Source Data file.

## Methods

**Study subjects**. As a part of the observational clinical trial (ChiCTR-PIC-17012358, http://www.chictr.org.cn/index.aspx, a summary (translated version) of the approved study protocol has been submitted as Supplementary Note 1), all clinical samples including peripheral blood, ascites fluid, and tissues were obtained from healthy donors and GC patients in the department of gastrointestinal surgery at Renji hospital from January 2019 to July 2019. While the GC cohort for clinicopathological characteristics and prognosis analysis were retrospectively collected from January 2010 to December 2015. All the patients were treated as per standard of care described in our previously study[6,45–47]. In brief, GC was diagnosed by endoscopy with histopathological examination and tumor staging was evaluated according to the 7th edition of the UICC TNM classification, including patients with clinical stage (cStage) I were prone to received laparoscopic gastrectomy, while patients with cStage II/III disease underwent open gastrectomy with radical gastrectomy + D2 lymphadenectomy. Then adjuvant chemotherapy was recommended to pathological stage (pStage) II/III patients. After discharge, the patients were followed up by outpatient or phone calls with every 6 months until January 2021 or death. OS was defined as the duration from operation date to death date. RFS was defined as the duration from operation date to tumor recurrence or metastasis date.

The Ethics Committees of Renji Hospital affiliated to Shanghai Jiao Tong University School of Medicine approved the study protocols (including healthy donors and GC patients), and written informed consent was obtained from all subjects in this study. All the research was carried out in accordance with the provisions of the Helsinki Declaration of 1975.

**Postoperative complication evaluation**. Postoperative complications were monitored during postoperative management and only those occurring within 1 month after operation were recorded as an event. Firstly, postoperative complication was evaluated by Clavien-Dindo classification[48,49] according to treatment. Afterwards, a complications list proposed by Gastrectomy Complications Consensus Group[50] was used to define each complication, based on clinical complains, physical examination and radiology (X-ray, ultrasonography and computed tomography (CT)) reports. For example, duodenal or anastomotic leak was diagnosed by the presence of abdominal pain, fever (body temperature ≥38.0 °C), saliva or gastrointestinal contents in the drain or during relaparotomy, and radiologically by the contrast swallow test[51]; Abdominal infection or abscess was diagnosed by the presence of abdominal pain, fever (body temperature ≥38.0 °C), pus in the drain or that abdominal fluid collection or abscess were found on CT or during relaparotomy[52]. Those two above postoperative complications were categorized as AIC. Pneumonia was defined as a newly developed infiltrates on the chest radiograph and positive results of bronchoalveolar lavage culture[12,53,54]. Other complications including delayed gastric emptying or bowel obstruction requiring parenteral nutrition, pleural effusion requiring drainage, postoperative bleeding requiring both urgent transfusions and invasive treatment, myocardial infarction with patient's transfer to CCU/ICU/other critical care facility et al. were categorized as Non-AIC.

**Peripheral blood and abdominal ascites analysis**. Venous blood samples were obtained from healthy donors, preoperative GC (control group), postoperative GC without AIC (Non-AIC group, obtained on postoperative day 5) and postoperative GC with AIC (AIC group, obtained on AIC diagnosis day). Preoperative ascites fluid was collected before GC resection during operation. Then postoperative ascites fluid was collected by drainage tube on postoperative day 5 for non-AIC group and on AIC diagnosis day for AIC group. Neutrophils counts in ascites fluid were quantified by SYSMEX XNL-350 in the department of Laboratory Medicine. Briefly, anticoagulant tubes were used to collecting ascites fluid by drainage tube, and the test was completed within 4 h after collection in strict accordance with the SYSMEX XNL-350 instructions.

**ELISA assay**. We detected serum, plasma and ascites fluid MPO–DNA using a previously described sandwich ELISA method[55]. Briefly, 96-well microtiter plates were coated with 5 μg/ml anti-MPO monoclonal antibody (R&D, AF3667) as the capturing antibody overnight at 4 °C. After blocking by incubation buffer for 2 h, 45 μl samples were added per well in combination with Quant-iTTM Pico-GreenTM dsDNA (ThermoFisher, P7589) following the manufacturer's instructions. Absorbance was measured at a wavelength of 405 nm with a Synergy HT Multi-Mode Micro-plate Reader (BioTek). After incubation at 37 °C for 40 min, the optical density was measured. The serum or ascites fluid TGF-β1 (abcam, ab100647; MultiSciences, 70-EK981-96) or IL-8 (MultiSciences, 70-EK108HS-96) concentrations were also quantitated by ELISA following the manufacturer's instructions. All the values were determined spectrophotometrically by the absorbance at 450 nm using a microplate reader (BioTeK, Epoch).

**Neutrophil isolation and NETs purification**. Human whole blood and ascites fluid samples were collected into sterile vacutainers with ethylene diamine tetra-acetic acid (EDTA) and then were isolated for neutrophils by high-density gradient centrifugation. Samples was layered on Ficoll-Paque PLUS (GE Healthcare, 17-1440-02/17144002) and centrifuged at 1000 g for 30 min at room temperature. After discarding the supernatant fluid, the neutrophil population was separated by treating with lysing buffer to eliminate erythrocytes. Isolated neutrophils were resuspended in RPMI 1640 medium containing 0.5% serum and 1% penicillin/streptomycin (P/S). The viability of neutrophils was checked by flow cytometry (detected in BD Fortessa FACS with FACSDiva software v6.0) and it always was >95% (analyzed using FlowJo Software version 10.4.2 software). For NETs purification, neutrophils supernatant was centrifuged at 18,000 × $g$ for 10 min at 4 °C. Then the pellet was resuspended in 100 μl of cold PBS for sandwich ELISA assay and SEM.

**Immunofluorescence staining**. Tissue was firstly fixed in 4% paraformaldehyde at 4 °C, washed with PBS twice, embedded in paraffin and sectioned at 5 μm thickness. Then those paraffin-embedded tissue sections were deparaffinized, rehydrated and antigen retrieved in EDTA buffer. Sections were blocked with Fc Receptor blocker (Innovex, IMI01572E), and blocking with 5% BSA for 25 min at room temperature. Isolated neutrophils from indicated groups were firstly incubated for 3 h without any treatment or exposed to DNase I (0.25 mg/ml, Roche, 11284932001) or GC cells (1 × 10^5 MKN-45, MGC-803 or AGS) in vitro. Afterwards, we fixed these neutrophils on poly-L-lysine-coated coverslips (Corning, 354085) with 4% paraformaldehyde for 15 min at room temperature, washed with PBS twice, and permeabilized with 0.5% Triton X-100 (Beyotime, P0096) for 3 min. GC cells pretreated with medium containing indicated neutrophils from indicated groups, DNase I (0.25 mg/ml, Roche, 11284932001), PAD4i (10 μM, abcam, ab223598), NEi (5 μM, abcam, ab142369) for 1 day (replacing medium every 8 h) were fixed with 4% paraformaldehyde for 15 min at room temperature, washed with PBS twice, and permeabilized with 0.5% Triton X-100 (Beyotime, P0096) for 10 min. Then those samples were following incubated with anti-Cit-H3 (1:100, Abcam, ab5103), anti-MPO (10 μg/ml, R&D, AF3667), anti-TGF-β1(1:100, Thermo Fisher, MA1-21595; 1:200, Servicebio, GB14154), anti-E-cadherin (1:400, Servicebio, GB11082), anti-N-cadherin (1:500, Servicebio, GB12135) or anti-p-Smad2 (1:300, Thermo Fisher, 44-244 G) overnight at 4 °C. Next, fluorochrome-conjugated secondary antibodies (1:400, Invitrogen; 1:500, Servicebio) were added for incubation 30 min. After rinsing twice by PBS, DAPI (2 μg/ml, Servicebio, G1012), Hochest 33258 (1 μg/ml, Thermo Fisher, H1398) or Sytox green (1 μM, Thermo Fisher, R37109) was then used for counterstaining the nuclei and images were obtained by Leica DMI6000 inverted microscope or Microscope slide scanner (Pannoramic MIDI:3DHISTECH) and then analyzed by photoshop 5.0. NETs quantification in immunofluorescence referred to Albrengues et al.'s study[19], in which the NETs were counted as extracellular citrullinated histone-3-positive cells at least three representative immunofluorescence images (from two neighboring sections) per sample. Then neutrophils were counted as MPO-positive cells at the same images. The percentage of NETs-forming neutrophils was calculated using the formula: (number of NETs-forming neutrophils/number of neutrophils)*100. All values were determined by two pathologists who were blind to clinical or experimental information.

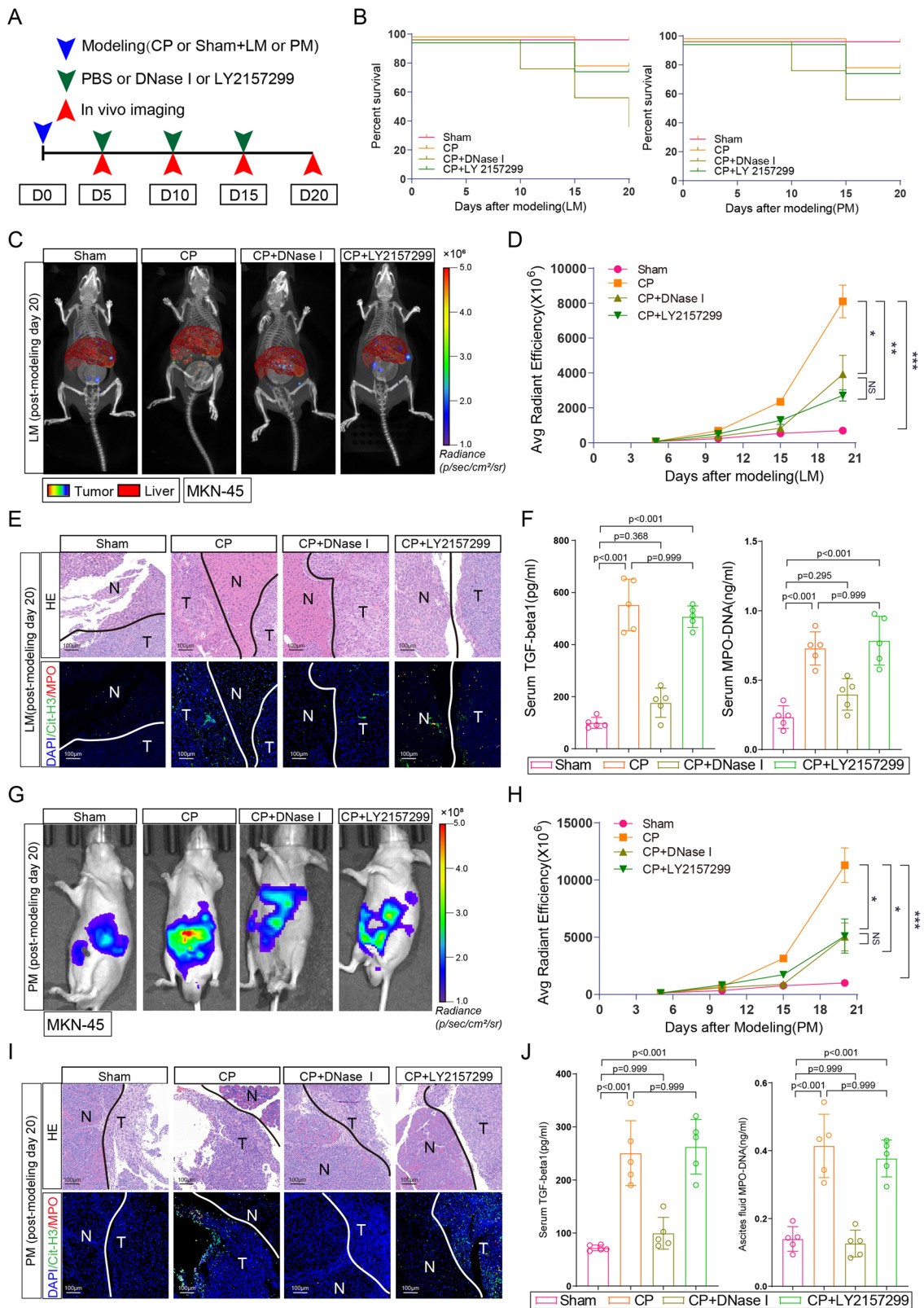

**Scanning electron microscopy (SEM).** Isolated neutrophils or its supernatant from indicated groups including GC patients or nude mice were plated on poly-L-lysine-coated coverslips and fixed in 2% glutaraldehyde and then post-fixed using repeat incubations with 1% osmium tetroxide/1% tannic acid. Samples were dehydrated with graded increasing concentrations of ethanol, and coated with 5 nm carbon. Imaging was taken by the Hitachi TEM system. For NETs-GC clusters quantification in SEM, it was determined by the mean percentage of the neutrophils (white arrow)

releasing webs (green arrow) binding with GC cells (red arrow) in at least ten representative views.

**Immunohistochemistry of tissues samples.** Samples from GC patients or nude mice with liver or peritoneal metastasis were resected and fixed in 10% formalin for 24 h, dehydrated and embedded in paraffin and then sectioned with a microtome (Leica, Deerfield, IL). Immunohistochemistry (IHC) was performed on dewaxed

**Fig. 6 TGF-β inhibitor effectively counteracts NETs-GC clusters induced metastasis but not aggravated sepsis. A** Schematic illustrating the experiment design; **B** Survival rate at 20 days in LM (left panel) and PM (right panel) nude mice as indicated modeling (n = 5 per group); **C, D** Representative images (**C**) and quantification (**D**) of metastatic lesions in LM nude mice with the indicated treatments; **E** Representative HE and immunofluorescence co-staining images of DAPI, Cit-H3, and MPO to assess NETs in the liver of LM nude mice at post-modeling day 20 (n = 5 per group). T = Tumor, N = Normal; **F** Serum TGF-β1 and MPO–DNA levels in LM nude mice at post-modeling day 20; **G, H** Representative images (**G**) and quantification (**H**) of metastatic lesions in PM nude mice with the indicated treatments; **I** Representative HE and immunofluorescence co-staining images of DAPI, Cit-H3, and MPO to assess NETs in the peritoneum of PM nude mice at post-modeling day 20. T = Tumor, N = Normal (n = 5 per group); **J** Ascites fluid TGF-β1 and MPO–DNA levels in PM nude mice at post-modeling day 20. Data represent the mean ± S.D. in **D, F, H, J** (n = 5 per group), one-way ANOVA with Tukey test was used in **D, F, H, J**. Source data are provided as a Source Data file.

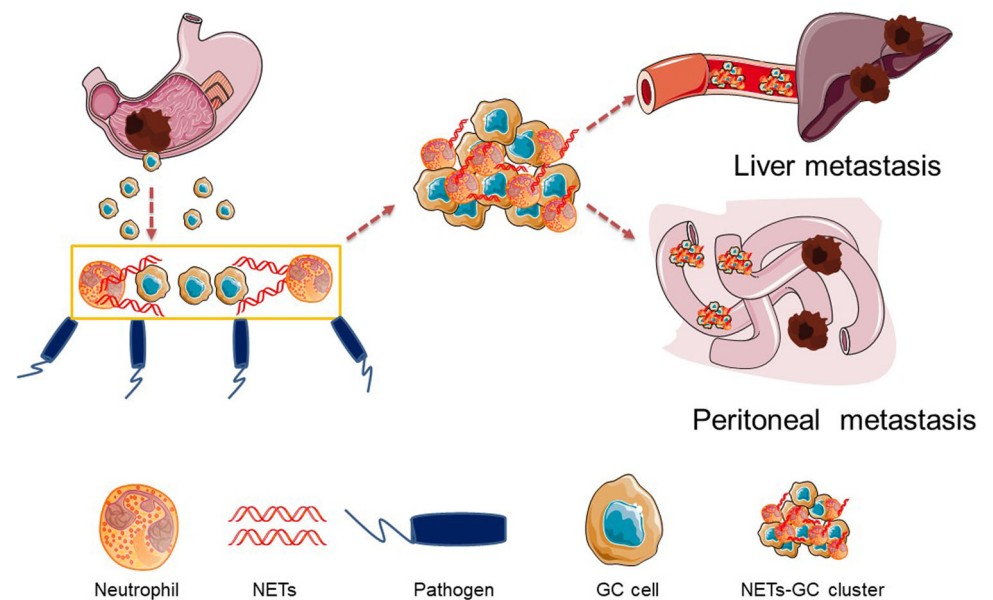

**Fig. 7 Proposed mechanism for the metastatic tropism of NETs-GC clusters due to postoperative AIC.** Free GC cells from primary GC could develop into CGCs (circulating GC cells) in peripheral blood or DGCs (disseminated GC cells) in abdominal cavity. Once postoperative AIC occur after surgery, pathogens or inflammatory factors would stimulate neutrophils to release NETs not only in peripheral blood but also at infectious sites (abdominal cavity), then in which those NETs trap CGCs or DGCs to form NETs-GC clusters, that potentially facilitate GC cells metastasis.

and hydrated 4-mm-thick sections of tissue using E-cadherin (1:800, Servicebio, GB11082), N-cadherin(1:800, Servicebio, GB11135) and Ki67(1:800, Servicebio, GB111499). After blocking with 1% BSA, the sections were incubated overnight at 4 °C with primary antibody followed by incubation with HRP-conjugated secondary antibody for 2 h at room temperature. The intensity of Ki67 expression was evaluated at least three representative images (from two neighboring sections) per sample by two independent pathologists with the formula: (number of ki67-positive GC cells/ all GC cells)*100.

**Transwell assay**. For GC cells transwell assay($5 \times 10^4$ for invasion assay or $2 \times 10^4$ for migration assay, Fig. 2A and Supplementary Fig. 3C), GC cells were plated on the upper wells with or without Matrigel-coated membrane for 6 h. Then after GC cells adherence, the medium was replaced by 150 μl conditioned medium such as indicated neutrophils ($1 \times 10^5$) from indicated groups, DNase I (0.25 mg/ml, Roche, 11284932001), PAD4i (10 μM, abcam, ab223598), NEi (5 μM, abcam, ab142369), TGF-β1 (0.2 ng/ml, MedChemExpress, P01137) or LY 2157299 (5 μM, Selleck, S2704) and incubated for 8 h. For NETs-GC clusters transwell assay, GC cells ($5 \times 10^4$ for invasion assay or $2 \times 10^4$ for migration assay, Fig. 5A–D) mixed with indicated neutrophils ($1 \times 10^5$), DNase I (0.25 mg/ml) or LY 2157299 (5 μM) were seeded on the upper wells with or without Matrigel-coated membrane for 8 h. For cells counting thereafter, the penetrated GC cells from the upper surface of the membrane were wiped off, and penetrated cells that crossed the membrane were fixed with 4% paraformaldehyde and stained with crystal violet. The number of penetrated GC cells (flat and irregular morphology) were counted under a light microscope in three fields of view, and the average number of cells was calculated.

**Western blot analysis**. Immunoblotting was performed using standard procedures. Total or nucleus protein were extracted using RIPA Lysis and Extraction Buffer (Thermo Scientific, 89901) or Nuclear and Cytoplasmic Extraction Reagents

(Thermo Scientific, 78833) according to per manufacturer's instructions. Briefly, equivalent amounts of protein were separated by sodium dodecyl sulfate(SDS)-polyacrylamide gel electrophoresis (PAGE) at 80 V for 2 h and transfected to polyvinylidene fluoride (PVDF) membranes for 2.5 h. Then the membranes were washed using 1% tris buffered saline Tween (TBST) for 25 min after incubation with specific antibodies targeting E-cadherin (1:2000, Abcam, ab40772), N-cadherin (1:2000, Abcam, ab18203), p-Smad2 (1:2000, Abcam, ab53100), Smad2 (1:2000, Abcam, ab40855), p-Smad3 (1:2000, Abcam, ab52903), Smad3 (1:2000, Abcam, ab84177), Lamin B1 (1:2000, Abcam, ab133741) at 4 °C overnight, then incubated with secondary antibodies labeled with horseradish peroxidase for 2 h. Finally, the membranes were detected by chemiluminescence system (BIO-RAD) and β-actin was used as an internal control. All immunoblotting original uncropped and unprocessed images were provided in Source data file.

**Cell-viability assay**. GC cells were seeded in 96-well plates at a density of 5000 cells per well. After incubation for 4 h (GC cells adherence), the medium was replaced by 100 μl conditioned medium such as PBS, indicated neutrophils ($1 \times 10^5$), DNase I (0.25 mg/ml, Roche, 11284932001) or neutrophils supernatant and then the medium was changed every 8 h. At indicated times thereafter, culture mediums were removed and then after gently washing with PBS three times, 10% CCK-8 (Dojindo, C0038) with RPMI 1640 were added to incubate for a further 1 h. Finally, cell viability was assessed at a wavelength of 450 nm after CCK-8 addition. Three independent experiments were carried out.

**RNA isolation and quantitative RT-PCR**. RNA was extracted from cultured cells with TRIzol reagent (Invitrogen) according to the manufacturer's instructions. c DNA was synthesized with ImProm-II Reverse Transcription System (Promega, Madison, USA). RT-PCR was performed with SYBR Green PCR Mater Mixture Reagents (Applied Biosystems, Carlsbad, USA) on the ABI 7300 PCR system

(Applied Biosystems). Data analysis of mRNA was normalized to the internal control β-actin and evaluated using the $2^{-\Delta\Delta Ct}$ method. Paired primers for indicated genes in this study were described below. E-cadherin Forward: GTG TTC GCT ATT GGA CGG GA; Reverse: TCA TAA CAG CCG TAC CTG GC. N-cadherin Forward: CTG TCT GGA AAA CAC CGA GC; Reverse: TTT CTG CTC CCG CCA CAA A.

**Animals experiments.** The nude mice experiments were approved by Renji Hospital Animal Care and Use Committee. Eight-week-old female athymic BALB/c nude mice were purchased from Shanghai SLAC Laboratory Animal Co., Ltd. with a weight between 20 and 25 g. For in vivo LM and PM models (Fig. 2), GC cells firstly seeded at the 10 cm culture dishes and then cocultured with medium containing indicated neutrophils from indicated groups, DNase I (0.25 mg/ml, Roche, 11284932001), PAD4i (10 μM, abcam, ab223598), NEi (5 μM, abcam, ab142369) for 3 days (replacing medium every 8 h). Afterwards, those above $5 \times 10^6$ GC cells were injected to the spleen or abdominal cavity under general anesthesia for LM and PM models, as described in our previous study[56,57]. The luciferase signal intensity was examined every 5 days after injection. For abdominal infectious nude mice models (Supplementary Fig. 5), we referred to cecal ligation puncture (CLP) model[27]. In brief, after anesthetizing and laparotomy, the cecum with (CLP) or without ligation (CP) was perforated with 21G or 25G needle. Then the cecum was returned to the abdominal cavity and the skin was closed. For CP or Sham with LM or PM models, CP or Sham operation with GC cells or GFP-GC cells injected to the spleen or abdominal cavity were performed at day 0. Then treatment including PBS or DNase I (2.5 mg/kg, Roche, 11284932001) or LY 2157299 (300 μg/mouse, Selleck, S2704) were administrated intraperitoneally every 5 days. CT-combined 3D organ reconstruction bioluminescence for LM as well as bioluminescence for PM were used for GC metastasis observation and comparison every 5 days until 20 days and the detailed operative process had been described[56,57]. Meanwhile, a clinical score for monitoring the infectious degree of nude mice was used[58]. The following each sign including lethargy, piloerection, tremors, periorbital exudates, respiratory distress, or diarrhea were noted as one point while every symptom was only assessed for its presence other than severity. Metastatic progression was monitored and quantified using a in vivo imaging system Spectrum (Caliper Life Sciences, Waltham, MA, USA), where firefly bioluminescence signals detection merged with CT image after intraperitoneal injection of 150 mg D-luciferin (Promega) at volume of 200 μl. All these mice were killed after luciferase signal intensity examination and then the blood, liver or peritoneum samples were subjected to ELISA, H&E staining, IHC, IF or SEM assays. The maximally permitted tumor diameter of 20 mm in any dimension was never exceeded

**Statistics.** Data were examined whether they were normally distributed with the D'Agostino and Pearson omnibus normality test. For normal distributed data, Student's $t$-test was applied to compare the difference between two groups or one-way analysis of variance (ANOVA) test was applied to compare the difference among three or more groups. For survival analysis, Kaplan–Meier method and log-rank test were used to compare RFS and OS. $p$-values were adjusted for multiple comparisons using the Bonferroni correction. Pearson correlation coefficient were used to analyzed the correlation of NETs (MPO–DNA concentration) and TGF-β1 level. All statistical calculation was performed using SPSS software package (version 23.0, IBM SPSS) or GraphPad Prism software versions 8.0. For the multiple comparisons of RFS and OS rate, $p < 0.017$ (0.05/3) was defined as statistically significant. For other tests, $p < 0.05$ was considered to indicate a statistically significant difference. All $p$-values were two-sided unless otherwise specified.

**Reporting summary.** Further information on research design is available in the Nature Research Reporting Summary linked to this article.

## Data availability
Source data for Fig. 1A, B, D, F, Fig. 2A–D, Fig. 3A, B, D, Fig. 4A–F, Fig. 5B, C, D, Fig. 6D, F, H, I, Supplementary Fig. 1A, B, C, Supplementary Fig. 2D, E, Supplementary Fig. 3A, B, D, F, G, Supplementary Fig. 4C, D, E, Supplementary Fig. 5E, Supplementary Fig. 6A, B, C, D, F, Supplementary Fig. 7C, D, E are provided as Source Data file. The authors declare that all data supporting the findings of this study are available within the Article, Supplementary Information or Source Data file. Source data are provided with this paper.

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

## Acknowledgements

This work was supported by National Natural Science Foundation of China (81802313 (X.X.), 31872740 (G.Z.), and 32070878 (G.Z.)); Shanghai Municipal Education Commission—Gaofeng Clinical Medicine Grant Support (20191905 (G.Z.)). The funding bodies had no role in the design of the study and collection, analysis, and interpretation of data and in the writing of the manuscript.

## Author contributions

X.X., Z.Z.Z., C.C.Z., and S.C.W. made substantial contributions to the conception and design of the study, acquisition of data, and analysis and interpretation of data; F.R.Y. and S.F.Y. were involved in drafting the manuscript and revising it critically for important intellectual content; E.H.Z. and G.Z. made substantial contributions to acquisition of data, and analysis and interpretation of data.

## Competing interests

The authors declare no competing interests.
