## [Peer Review File · Nature Communications]

Reviewers' Comments:

Reviewer #1:

Remarks to the Author:

In this manuscript, Xia et al. use clinical samples and different experimental in vitro and in vivo models to propose that NETs, produced during infectious complications after gastrectomy, facilitate gastric cancer recurrence at metastatic sites. The authors aim to show that NETs can trap cancer cells and promote a TGF-beta dependent-EMT to facilitate metastasis. Interestingly, the authors used cecal ligation in mice to recapitulate abdominal infection and sepsis, which mimic postoperative infections following gastrectomy. In this model, while digesting NETs with DNase I aggravated infection, sepsis and increased death rate in mice, the use of a TGFb inhibitor to counteract NETs-induced EMT did not aggravated sepsis and was efficient at suppressing metastasis. The authors suggest that TGFb inhibition do not interfere with neutrophil recruitment and NET formation (therefore leaving antimicrobial defense of NETs intact), while counteracting NET pro-metastatic phenotype. Overall, the manuscript and hypothesis tested are interesting; however, it is not written clearly, lacks experimental controls and in most instances, data are over interpreted and do not support their conclusions. Moreover, in the last few years different groups have documented the effect of neutrophils and NETs in infection and inflammation-induced metastasis (El rayes, 2015; De Cock, 2016; Albregues, 2018; Krall, 2018).

Overall strengths: The hypothesis is interesting, important, and highlight a key area of work in the metastasis field. The clinical association between NETs and infectious complications in gastric cancer progression is novel. The use of neutrophils derived from healthy donor, and patients with and without postoperative infectious complications is also a strength of the study.

Overall weaknesses:

The manuscript is poorly written, and data are over-interpreted. Key control experiments are lacking to conclude properly the role of NETs in postoperative infection and recurrence. Therefore, it is unclear whether NETs are specifically mediating recurrence following postoperative infections, mainly because of the lack of methods to detect and inhibit NETs specifically. The authors use artificial migration and invasion assay in vitro to show the effect of NETs on cancer cells invasive behavior, however it is unclear whether this phenomenon is also observed in in vivo models. Indeed, it is possible that NETs could promote extravasation or proliferation of the cancer cells at metastatic sites as it has been shown before (these papers being cited in the manuscript: Cools-Lartigue, 2013; Tohme, 2016; Park, 2016; Albregues 2018; Lee, 2019). Overall this manuscript lacks major experimental controls to correctly interpret the data.

The work from Xia et al. is of interest and represent another step toward the understanding of NETs in cancer recurrence. However, I have concerns regarding this work which still lacks experimental evidence.

Major concerns:

1- NETs are not quantified and visualized properly in all the experiments presented. In vitro and in vivo NETs are commonly detected via the co-localization of DNA, Citrullinated Histone and at least one NET-associated protease (Neutrophil Elastase and Myeloperoxidase). In this manuscript, authors mainly use MPO as a marker of NETs, which is not an appropriate marker when used alone. Indeed, MPO is commonly expressed by all neutrophils, independently of their activation status. Moreover, in Fig. 1A, it is highly surprising to see what appears a few MPO positive cells and mostly MPO negative cells (weaker signal looks like background from cells auto-fluorescence). This raises the question of the purity of neutrophils used in this study. Moreover, the authors use independently, quantification of MPO, NE and DNA in the serum to look at NET markers. MPO and NE are thought to be released by activated neutrophils by 2 different mechanisms: 1- degranulation of free NE and MPO 2- the release of NET-associated MPO and NE. Therefore, NETs can be properly quantified in the serum via the use of a sandwich ELISA, using antibodies against the DNA, and antibodies against MPO or NE (protocols can be found in multiple papers on NET biology and their involvement in different inflammatory diseases).

2- The authors need to target NETs more specifically (only DNase I is used). This can be done by using PAD4 inhibitors and NE inhibitors, which are commonly used by different groups working on NET biology.

3- In Fig. 2B, C, E, F, it is impossible to conclude that NETs promote metastasis. The authors need to use NET-targeting strategies such as DNase I, NE inhibitor, and PAD4 inhibitor. It is possible that in these experiments non-NET-forming neutrophils are important. Indeed, it has been shown that direct contact between neutrophils and cancer cells in the circulation is sufficient to help the cancer cells to expand their metastatic potential (Szczerba, 2019).

4- The authors conclude from Figure 2 that NETs promote invasion and migration in vivo, but it is impossible to reach this conclusion based on the experiments presented in this manuscript. It is indeed hard to understand how a tail vein injection model relate to the invasive behavior of GC cells.

5- The cellular and molecular mechanisms of NET-induced recurrence is unclear. Indeed, while the authors claim that NETs can promote EMT in GC cells to enhance metastasis, it is unclear how EMT participate on metastasis in the mouse model used. It is therefore also unclear to conclude which part(s) of the metastatic process is enhanced by NETs in vivo and how it relates to their ability to promote EMT. Indeed, NETs could also play a role on extravasation and proliferation, as it has been previously published. For example, the authors use IV injection of GC cells to generate CTCs, while this method is commonly use by the scientific community to generate lung metastasis (the cancer cells will be trapped into the lungs). Therefore, it is impossible to conclude the proper effect of NETs in these experiments. The only conclusion possible is that NETs facilitate metastasis through unknown processes, which is already a brand-new discovery considering the context of post-operative complications. It is important here to quantify CTCs or GC cells already lodged at the metastatic sites (that have extravasated) at different time point in the experiments (for example the day after IV injection and one week after for example). It is indeed so far, impossible to conclude whether neutrophils/NETs can help CTCs to extravasate, to survive in the circulation (through direct contact with neutrophils), to survive at the metastatic site, or to proliferate at the metastatic site.

6- From the bioluminescence quantification, it is hard to believe that at day 0, after IV injection of the cancer cells, no signal can be detected (using this method, all cells should be viable and trapped in the lungs, and bioluminescence at day 0 is a key control of a well performed IV injection).

7- The authors mainly look at migration and invasion in vitro, however in vivo, from the experimental procedures used, it is hard to conclude if the invasive ability of the GC cells play a role in facilitating metastasis.

8- The authors suggest that NETs are released in peripheral blood in their different in vivo models, but it is impossible to reach this conclusion with the data presented. Indeed, NETs could be formed in the circulation, but also at the infectious site. If this is the latest explanation, it is hard to believe that NETs can therefore trap the cancer cells in the circulation to bring them to the infectious site where they would proliferate to form metastasis.

9- The authors only focus on neutrophils, while the Weinberg group (Krall, 2018) has linked surgical resection with recurrence through a systematic inflammatory response involving inflammatory monocytes macrophages and T cells. Of note, in their model, inflammatory monocytes were responsible for recurrence and not neutrophils. This needs to be assessed and discuss. In addition, the authors show that surgical stress can promote NET formation, even in the absence of postoperative infection, but these NETs do not seem to have pro-metastatic properties. Is it possible that in this context, inflammatory monocytes are more important than neutrophils in mediating metastasis?

10- The authors state that IL-8 helps to recruit neutrophils to infectious sites in their model, while they only show that IL-8 is present in the serum of IC patient (supp 3G). This is just a simple correlation. If the authors want to claim that IL-8 is involved in neutrophils chemotaxis in their model, they need to show it properly with the use of blocking antibodies or pharmacological inhibitors. This comment needs to be considered for most of the conclusion regarding the role of IL-8 in the authors model.

11- IL-8 is used as a chemotactic factor in Fig.5, however IL-8 has been reported to promote NET formation in multiple papers (Brinkmann, 2004; Keshari, 2012; An, 2019; Nie, 2019 – to name only a few). It is therefore intriguing to see that the presence of IL-8 do no have any effect on the invasion of cancer cells cultured with non-IC neutrophils, considering that NETs should be formed also in this condition (and this needs to be controlled). Another problem comes from Fig. 5B, where the basal invasion of GC cells is the same in all the condition. Indeed in Fig. 2A and 2D, there is a clear increase in the IC condition (Fig. 2A and 2D). The authors should explain this discrepancy.

12- In Fig. 5D, the authors use medium in the lower chamber for SEM. They need to explain why cells would be present in the medium and not on the lowest membrane of the Boyden chamber.

13- Some cancer cells can promote the formation of NETs (Park, 2016; Lee, 2019). The ability of the GC cells used in this manuscript to promote NET formation needs to be assessed, because it could alter all the conclusion from the authors.

14- EMT is assessed in vivo in Fig. 3E and 3F thanks to different epithelial and mesenchymal markers. However, the pattern exposed in these representative pictures seems unlikely. For example, non-IC lungs should have no to little metastasis (from previous experiments), and indeed, from the histology of the lungs presented, the lungs look quite normal. However, all the cells in the lungs seems to be E-cad positive, which seems unlikely. Even more surprisingly, in the IC group, metastases should be present, and the histology of the lungs look quite normal as well, without any apparent metastasis, however all the lung cells are now E-cadherin negative. One can ask why all normal cells are E-cad positive in the non-IC group and why all normal cells are E-cad negative in the IC group. The same comments apply to the mesenchymal markers (N-cad, vimentin, snail). The staining pattern needs to be addressed and more importantly EMT needs to be assessed specifically in the cancer cells. This can be done by staining the cancer cells (they express luciferase, and luciferase antibodies can be used to detect them).

15- TGF-beta needs to be quantified in the serum of IC and non-IC patients to test its correlation with the presence of NETs. TGF-beta also needs to be quantified in IC and non-IC neutrophils culture.

16- The authors show that neutrophils supernatant from IC neutrophils do not promote invasion and migration, and therefore conclude that direct contact between NET-forming neutrophils and GC cells is important. However, from the data presented, it is impossible to know if NETs are present in the supernatant (there should be some NETs as supernatant is commonly used by groups working on NET biology to assess the effect of "free NETs"). Moreover, NETs in the supernatant ("free NETs") have been shown to bind to cancer cells and this need to be assessed and discussed (Najmeh, 2017; Monti, 2017 and 2018 - to name a few). Interestingly, it has also been previously published that NET-supernatant can promote breast cancer cells invasion in vitro (Park, 2016) and therefore, the differences observed need to be discussed.

17- The authors show that IC-neutrophils supernatant does not promote invasion and migration of the cancer cells. It is important to test the presence of TGF-beta in the culture supernatants and its effect on EMT markers. Indeed, while all the experiments with TGF-beta inhibition are interesting, it requires more investigation. For example, is TGF-beta present on the NETs? Where is TGF-beta coming from?

18- The authors conclude from Figure 5 that after LPS treatment NETs can carry the cancer cells to

infectious site through chemotaxis. There are mainly two problems on this conclusion. First, LPS is administered 5 days after cancer cell injection, and it is not clear whether at this point, CTCs are presents or if the cancer cells extravasated in the lungs. Therefore, it is impossible to conclude whether NETs trap the cancer cells in the circulation to bring them to the metastatic site (and it is impossible to know where the NETs are form, in the circulation or at the infectious site). Second, the authors do not assess properly chemotaxis of cancer cells at the infectious site considering that tail vein injection is used, which commonly leads to cancer cells being trapped in the lungs.

19- To test whether the NETs really trap the cancer cells to infectious site, the presence of CTCs needs to be quantified and a proper extravasation assay needs to be performed. Also, NETs have been shown to promote proliferation of low cycling cancer cells in the lungs (Albregues, 2018). The authors seem to indicate that NET do not alter proliferation of the cancer cells in vitro, but in vivo, the environment is different, and NETs could play a role. Therefore, it would be interesting to digest NETs with DNase I at later time points to assess NETs effect on proliferation at metastatic sites.

20- In this manuscript, IC is defined by a combination of clinical findings and results of examinations. Is it possible to be more specific and to score these criteria to highlight the threshold between IC and non-IC patients?

Minor concerns:

1- Some sentences are not grammatically correct within the manuscript.

2- Some acronyms need to be spell out properly (for example: R0 resection, TNFa, IL-8, CRP and PCT).

3- Within the introduction, the authors states that solid data exploring NETs-directed therapy for clinical practice is absent, which is not true. See the papers that the authors are citing (Cools-Lartigue, 2013; Tohme, 2016; Park, 2016; Albregues 2018; Lee, 2019)

4- The authors collect peripheral blood samples from patients with or without postoperative ICs. It would be interesting to know which kind of ICs were involved (anastomotic leaks or pneumonia) and whether the isolated neutrophils are different, at least in their ability to form NETs.

5- Interestingly, elevated CRP and SAA were also associated with reduced disease-free survival for breast cancer patients. This paper should be cited as it seems to indicate a general mechanism in different type of solid cancer.

6- In fig. 1E, the number of neutrophils is quantified, but it is not written how. Did the authors used FACs and Ly6G staining, cells morphology, something else?

7- In fig. 1G, NETs are quantified, but it not written how. Considering that a major problem of the manuscript is the detection of NETs, this need to be clearly stated.

8- The authors mention using a TGF-B1 agonist which never seems to be used.

9- TGF-beta have been shown to modulate neutrophils phenotype (Fridlender, 2009) toward a "N1" or "N2" phenotype. The authors should assess the effect of TGF-beta of the ability to induce or inhibit NETs in IC and non-IC neutrophils.

10- A clinical score of sepsis is quantified in Fig 6B and supp 6C, but the method for such a quantification do not appear in the manuscript.

11- NETs digestion leads to sepsis in the CP model indicating that NETs are critical in eliminating pathogens. However, neutrophils can also phagocytose and degranulate to counteract an infection, and it might be important to analyze their ability to do so in the context of a DNase I treatment and TGF-beta inhibition.

12- The authors show that NETs can promote wound healing of cancer cells in vitro. However, it has been previously shown that NETs can counteract wound healing in patients with diabetes (Ling Wong, 2015). This needs to be discussed.

13- IL-8, TNF- α and CRP detection were performed in hospital's routine laboratory. The authors need to be more specific and provide protocol and antibody catalog numbers.

14- The concentration of DNase I used is not written for in vitro experiment.

15- For migration and invasion assay, the methods indicate a 8 μ m porous membrane, while Supp. Fig. 4 indicate a 4 μ m porous membrane.

16- When the authors are quantifying migration/invasion of the cancer cells, are they using morphological differences to distinguish between invading cancer cells and invading neutrophils?

Reviewer #2:

Remarks to the Author:

The manuscript "Neutrophil extracellular traps produced during postoperative infection facilitate recurrence of gastric cancer" investigated the mechanisms of postoperative infectious complications promote gastric cancer recurrence and decrease long-term survival. There were several comments on this manuscript:

1. The most important are the survival of infectious complication affect the patients' survival. Patients with complications usually with more advanced cancer, so the comparison of patients with or without complication need balance the clinicopathologic data first, especially for stages.
2. The figures were so small and difficult to see the figures well.
3. There was not too much innovations.

Reviewer #3:

Remarks to the Author:

In the manuscript of Xia and colleagues the authors seem to present a consistent story using a variety of in vitro and preclinical in vivo model to draw a functional link between infectious complication-associated neutrophil NET formation and dissemination of gastric cancer cell lines. Such observations would be consistent that infectious complication associated with gastric cancer resection decreased survival.

The reason why my above statement is highly hypothetical results from the insurmountable challenge for this reviewer to deal with highly inappropriate small sizing of the data figures combined with their low resolution, which the authors chose for their submission. This prevents this assessor from evaluating the authors' interpretation of their data presented and hence to assess the extent by which the authors' texts appropriately reflect the data shown. In addition, this reviewer is also not in a position to assess the quality of the data presented in the first instance.

Notwithstanding my major critique above, as a minor comment, this author criticizes the authors for not having included scale bars in their photomicrographs.

Reviewers' comments:

Reviewer #1 (Remarks to the Author):

In this manuscript, Xia et al. use clinical samples and different experimental in vitro and in vivo models to propose that NETs, produced during infectious complications after gastrectomy, facilitate gastric cancer(GC) recurrence at metastatic sites. The authors aim to show that NETs can trap cancer cells and promote a TGF-beta dependent-EMT (epithelial-mesenchymal transition) to facilitate metastasis. Interestingly, the authors used cecal ligation in mice to recapitulate abdominal infection and sepsis, which mimic postoperative infections following gastrectomy. In this model, while digesting NETs with DNase I aggravated infection, sepsis and increased death rate in mice, the use of a TGFb inhibitor to counteract NETs-induced EMT did not aggravate sepsis and was efficient at suppressing metastasis. The authors suggest that TGFb inhibition do not interfere with neutrophil recruitment and NET formation (therefore leaving antimicrobial defense of NETs intact), while counteracting NET pro-metastatic phenotype. Overall, the manuscript and hypothesis tested are interesting; however, it is not written clearly, lacks experimental controls and in most instances, data are over interpreted and do not support their conclusions. Moreover, in the last few years different groups have documented the effect of neutrophils and NETs in infection and inflammation-induced metastasis (El rayes, 2015; De Cock, 2016; Albregues, 2018; Krall, 2018).

Overall strengths: The hypothesis is interesting, important, and highlight a key area of work in the metastasis field. The clinical association between NETs and infectious complications in GC progression is novel. The use of neutrophils derived from healthy donor, and patients with and without postoperative infectious complications is also a strength of the study.

Response Thank you for your positive and encouraging comments. In this revised manuscript, we had polished our article, added requested experiments and prudently interpreted our results. More importantly, in the process of revision according to your suggestions, we further enriched our proposed mechanism and theory for NETs-GC clusters induced GC metastasis.

Overall weaknesses:

The manuscript is poorly written, and data are over-interpreted. Key control experiments are lacking to conclude properly the role of NETs in postoperative infection and recurrence. Therefore, it is unclear whether NETs are specifically mediating recurrence following postoperative infections, mainly because of the lack of methods to detect and inhibit NETs specifically. The authors use artificial migration and invasion assay in vitro to show the effect of NETs on cancer cells invasive behavior, however it is unclear whether this phenomenon is also observed in in vivo models. Indeed, it is possible that NETs could promote extravasation or proliferation of the cancer cells at metastatic sites as it has been shown before (these papers being cited in the manuscript: Cools-Lartigue, 2013; Tohme, 2016; Park, 2016; Albregues 2018; Lee, 2019). Overall this manuscript lacks major experimental controls to correctly interpret the data.

The work from Xia et al. is of interest and represent another step toward the understanding of

NETs in cancer recurrence. However, I have concerns regarding this work which still lacks experimental evidence.

Response Thank you for your suggestions. We had invited a professional English editor to polish this manuscript. Meanwhile, several key control experiments including: 1) targeting NETs more specifically with PAD4 inhibitor and NE inhibitor; 2) replacing lung metastasis model with liver metastasis (LM) model in which GC cells invasive behavior could be investigated; 3) proliferation and extravasation experiments were designed and performed to interpret the roles of NETs in postoperative AIC and GC metastasis. We hope our added experimental evidences could consolidate our conclusion.

Major concerns:

1- NETs are not quantified and visualized properly in all the experiments presented. In vitro and in vivo NETs are commonly detected via the co-localization of DNA, Citrullinated Histone and at least one NET-associated protease (Neutrophil Elastase and Myeloperoxidase). In this manuscript, authors mainly use MPO as a marker of NETs, which is not an appropriate marker when used alone. Indeed, MPO is commonly expressed by all neutrophils, independently of their activation status. Moreover, in Fig. 1A, it is highly surprising to see what appears a few MPO positive cells and mostly MPO negative cells (weaker signal looks like background from cells auto-fluorescence). This raises the question of the purity of neutrophils used in this study. Moreover, the authors use independently, quantification of MPO, NE and DNA in the serum to look at NET markers. MPO and NE are thought to be released by activated neutrophils by 2 different mechanisms: 1- degranulation of free NE and MPO 2- the release of NET-associated MPO and NE. Therefore, NETs can be properly quantified in the serum via the use of a sandwich ELISA, using antibodies against the DNA, and antibodies against MPO or NE (protocols can be found in multiple papers on NET biology and their involvement in different inflammatory diseases).

Response Thank you for these constructive concerns about the NETs quantification and visualization. Your proposed experiments are absolutely key evidence and results to our conclusion. According to your suggestions and provided references {Yang, 2020 #678} {Lee, 2019 #86} {Albregues, 2018 #85} {Park, 2016 #324}, all NETs in our study were re-detected by immunofluorescence via the co-localization of DNA, Citrullinated Histone 3(Cit-H3) and Myeloperoxidase(MPO) *in vivo* and *in vitro*. Secondly, we also used the sandwich ELISA method as you suggested {Yang, 2020 #678} {Kessenbrock, 2009 #734} to quantified NETs in our study. Again, thank you for this insightful advisement.

2- The authors need to target NETs more specifically (only DNase I is used). This can be done by using PAD4 inhibitors and NE inhibitors, which are commonly used by different groups working on NET biology.

Response Thank you for your kind suggestions. We had redesigned our experiments and used different kinds of inhibitors, such as PAD4 inhibitor and NE inhibitor to target NETs biology more specifically both *in vivo* and *in vitro* (figure 2 and its result, yellow highlight). Those specifical

inhibitors used in our study help to consolidate our conclusion. Thank you.

3- In Fig. 2B, C, E, F, it is impossible to conclude that NETs promote metastasis. The authors need to use NET-targeting strategies such as DNase I, NE inhibitor, and PAD4 inhibitor. It is possible that in these experiments non-NET-forming neutrophils are important. Indeed, it has been shown that direct contact between neutrophils and cancer cells in the circulation is sufficient to help the cancer cells to expand their metastatic potential (Szczerba, 2019).

Response Thank you. As mentioned in “Major Concern 2”, we have used NETs-targeting strategies including DNase I, NE inhibitor and PAD4 inhibitor to consolidate our conclusion that NETs could promote metastasis. Still, the effects of non-NETs-forming neutrophils on GC cells were not significantly different among control, Non-AIC and NETs digestion groups. Furthermore, NETs-targeting strategies significantly abolished NETs’ effects on GC cells *in vivo* (figure 2 and its result). While in Szczerba et al.’s study {Szczerba, 2019 #738}, they found neutrophils can directly contact circulating tumor cells (CTCs) to form CTC-neutrophil clusters that representing key vulnerabilities of the metastatic process. Of note, they did not examine NETs expression in those neutrophils which directly contacted with CTCs. It is possible that those neutrophils (directly contacted with CTCs) formed NETs and then trapped breast cancer cells to travel in the circulation for metastasis, same as NETs-GC clusters proposed in our study (figure 7). Indeed, it has been reported that coculturing breast cancer cells with neutrophils could stimulate neutrophils to release NETs {Lee, 2019 #86} {Park, 2016 #324}. But in our GC study, coculturing GC cells with neutrophils failed to observe NETs formation (supplement figure 3 and its result), which subsequently inferred: 1) little NETs-GC clusters formation in this condition; 2) non-NETs-formation neutrophils had little effect on the metastatic ability of GC cells. Therefore, our results supported that NETs-formation neutrophils would trap free GC cells to form NETs-GC clusters and in the status, NETs could facilitate GC cells metastasis but Non-NETs-formation neutrophils did not. Still, thank you for your questions and we had added above mentioned opinion in our discussion part with yellow highlight.

4- The authors conclude from Figure 2 that NETs promote invasion and migration *in vivo*, but it is impossible to reach this conclusion based on the experiments presented in this manuscript. It is indeed hard to understand how a tail vein injection model relate to the invasive behavior of GC cells.

Response Thank you for raising concern. We have fully realized that our tail injection inducing lung metastasis model was not proper for invasion and migration research. Thus, we introduced intrasplenic injection inducing LM model to replace lung metastasis in our study. The main reasons include: 1) The implantation and outgrowth of CTCs in liver depend on the behaviors of both invasion and migration of GC cells. While a tail vein injection model in which CTCs just would be trapped in lung to generate metastatic niches was not appropriate enough, as you have pointed out; 2) In this model, injected GC cells would travel through portal veins (mimic CTCs) to generate metastatic niches in liver which could simulate GC liver metastasis in human; 3) This liver metastasis model has been demonstrated in our previous study {Li, 2019 #362}; 4) We could trail and observe the outgrowth and formation of GC cells through CT combined 3D organ reconstruction bioluminescence imaging and furthermore, clinical liver metastasis tissues of GC were collected for validation experiments; 5) Peritoneum and liver were the top two metastatic

sites in GC patients {D'Angelica, 2004 #739} {Nashimoto, 2013 #740}. Therefore, it is more significant and necessary to use LM and peritoneal metastasis (PM) nude mice models for the GC biological behavior study affected by NETs. Finally, we removed these results about lung metastasis nude mice model in our revised manuscript in order to diminish ambiguous interpretations.

5- The cellular and molecular mechanisms of NET-induced recurrence is unclear. Indeed, while the authors claim that NETs can promote EMT in GC cells to enhance metastasis, it is unclear how EMT participate on metastasis in the mouse model used. It is therefore also unclear to conclude which part(s) of the metastatic process is enhanced by NETs in vivo and how it relates to their ability to promote EMT. Indeed, NETs could also play a role on extravasation and proliferation, as it has been previously published. For example, the authors use IV injection of GC cells to generate CTCs, while this method is commonly use by the scientific community to generate lung metastasis (the cancer cells will be trapped into the lungs). Therefore, it is impossible to conclude the proper effect of NETs in these experiments. The only conclusion possible is that NETs facilitate metastasis through unknown processes, which is already a brand-new discovery considering the context of post-operative complications. It is important here to quantify CTCs or GC cells already lodged at the metastatic sites (that have extravasated) at different time point in the experiments (for example the day after IV injection and one week after for example). It is indeed so far, impossible to conclude whether neutrophils/NETs can help CTCs to extravasate, to survive in the circulation (through direct contact with neutrophils), to survive at the metastatic site, or to proliferate at the metastatic site.

Response Thank you for pointing out these problems. The key viewpoint of our study is that postoperative AIC would stimulate neutrophils to release NETs both in peripheral blood and abdominal cavity (figure 1, 5 and their results), where NETs could trap free GC cells to form NETs-GC clusters that facilitate them metastasis through TGF- β signaling pathway activation (figure 4, 5, 6 and its result). But as you suggested, we modeled LM and PM nude mice with cecal puncture without ligation (CP) that stimulated neutrophils to release NETs for NETs-GC clusters formation in peripheral blood and abdominal cavity (figure 5, supplement figure 5 and their results). Then we sacrificed indicated mice at post-modeling day 5 to quantify GC cells that extravasated, survived in the circulation or at the metastatic site. As a result, NETs-GC clusters were capable to help GC cells to extravasate and to survive both in the circulation and at the metastatic site (figure 5 and its result, yellow highlight).

6- From the bioluminescence quantification, it is hard to believe that at day 0, after IV injection of the cancer cells, no signal can be detected (using this method, all cells should be viable and trapped in the lungs, and bioluminescence at day 0 is a key control of a well performed IV injection).

Response Thank you for mentioning us these problems. The missing signals were due to the excessively upregulated threshold of detecting area at early days of mice models. We have also been aware of them and carefully repeated a few experiments and properly replaced them.

7- The authors mainly look at migration and invasion in vitro, however in vivo, from the experimental procedures used, it is hard to conclude if the invasive ability of the GC cells play a

role in facilitating metastasis.

Response Thank you for your question. We have additionally designed LM nude mice model to investigate the invasive ability of GC cells in facilitating metastasis *in vivo*. The results are presented in figure 2, 5 and 6 in which NETs facilitated GC cells to extravasate as well as form metastatic lesions in liver.

8- The authors suggest that NETs are released in peripheral blood in their different *in vivo* models, but it is impossible to reach this conclusion with the data presented. Indeed, NETs could be formed in the circulation, but also at the infectious site. If this is the latest explanation, it is hard to believe that NETs can therefore trap the cancer cells in the circulation to bring them to the infectious site where they would proliferate to form metastasis.

Response Thank you for raising this concern. Our results did show that NETs could be detected not only in the circulation, but also at the infectious or metastatic site *in vivo* models (figure 5, supplement figure 5), similar to GC patients experiencing postoperative AIC (figure 1). So far, NETs formation at distant organ has been reported to 1) trapped circulating tumor cells (CTC){Cools-Lartigue, 2013 #311}; 2) awaken dormant cancer to develop into metastasis{Albregues, 2018 #85}; 3) acted as a chemotactic factor to attract cancer cells to target site for metastasis{Yang, 2020 #678}. Then in the circulation: 1) NETs level (serum or plasm) could be a predictor for metastasis{Yang, 2020 #678}; 2) CTC-neutrophil clusters, representing key vulnerabilities of the metastatic process{Szczerba, 2019 #738}. While *in vitro* experiments, Lee et al. found cancer cells could bind to NETs-positive neutrophils and proposed an explanation that the NETs-rich niche traps cancer cells{Lee, 2019 #86}. In our study, we designed a specific transwell chamber experiment and GC-bearing nude mice with infection models to investigate whether NETs could trap GC cells and then facilitate them invasion, migration and metastasis. As shown in figure 5, more GC cells penetrated to lower chamber when coculturing with NETs-formation neutrophils compared to that with Non-NETs-formation neutrophils or TGF- β inhibitor administration groups. Notably, the representative attachment or trapping images were visualized by SEM in figure 5 both *in vitro* and *in vivo*: more GC cells attached to the webs of NETs in the NETs-formation group (NETs-GC clusters) compared to those in Non-NETs formation groups. Furthermore, we demonstrated that NETs-GC clusters could be observed in peripheral blood (LM model) and ascites fluid (PM model) from GC-bearing nude mice with infection (figure 5). Therefore, our proposed mechanism concerning NETs promoted metastasis was that postoperative AIC would stimulate neutrophils to release NETs not only in peripheral blood (circulation) but also at infectious sites (or abdominal cavity), then in which those NETs trapped free GC cells to form NETs-GC clusters, that potentially facilitate GC cells extravasation and proliferation for metastasis. All in all, our results were to provide a better understanding for postoperative AIC inducing poor prognosis. Still, thank you for your question and we had added those explanation to our discussion part with yellow highlight.

9- The authors only focus on neutrophils, while the Weinberg group (Krall, 2018) has linked surgical resection with recurrence through a systematic inflammatory response involving inflammatory monocytes macrophages and T cells. Of note, in their model, inflammatory monocytes were responsible for recurrence and not neutrophils. This needs to be assessed and discuss. In addition, the authors show that surgical stress can promote NET formation, even in the

absence of postoperative infection, but these NETs do not seem to have pro-metastatic properties. Is it possible that in this context, inflammatory monocytes are more important than neutrophils in mediating metastasis?

Response Thank you for reminding us of these points. Tumor immune microenvironment (TIME) have been recognized as important factors for cancer metastasis {Binnewies, 2018 #742}. In the TIME, T cells, tumor associated macrophages (TAMs), neutrophils and other immune cells all had been proven significantly functional for metastasis. In Krall et al.'s study, they found surgery induced systematic inflammatory monocytes were responsible for recurrence but not neutrophils. One possible explanation is that in that case, surgery induced systematic inflammatory could not stimulate neutrophils to produce enough NETs as infection did, which had also been showed in our study that neutrophils isolated from Non-AIC group (postoperative GC patients) produce more NETs than that from control (preoperative GC patients) but without significant. However, once surgery {Tohme, 2016 #312} or other factors, such as smoking {Albregues, 2018 #85}, LPS {Yang, 2020 #678}, induced enough NETs, they played significant and important roles in tumor metastasis or recurrence. These experimental results were consistent with our *in vivo* findings that nude mice with CP operation produced more NETs than sham did (figure 5, 6, supplement 5, 6 and their results). Therefore, it is something possible that in the context of surgical stress stimulating a little NETs release, inflammatory monocytes are more important than neutrophils in mediating metastasis. Nevertheless, in the context of enough NETs release, such as postoperative AIC stimulation, NETs do have significantly pro-metastatic properties. Thank you.

10- The authors state that IL-8 helps to recruit neutrophils to infectious sites in their model, while they only show that IL-8 is present in the serum of IC patient (supp 3G). This is just a simple correlation. If the authors want to claim that IL-8 is involved in neutrophils chemotaxis in their model, they need to show it properly with the use of blocking antibodies or pharmacological inhibitors. This comment needs to be considered for most of the conclusion regarding the role of IL-8 in the authors model.

Response Thank you for your kind suggestions. IL-8, alternatively known as CXCL8, was described as a potent chemotactic factor {Van Damme, 1988 #771} {Walz, 1987 #772} and NETs stimulus {Brinkmann, 2004 #258} {Keklikoglou, 2019 #486} for neutrophils. Then a growing number of recent studies reported that IL-8 could promote cancer progression via pro-EMT (epithelial-mesenchymal transition) or pro-angiogenic effects. Besides, several other chemokines (CXCL1, CXCL2, CXCL3, CXCL5, CXCL6, CXCL7) has also been reported to play crucial roles in combating microbial infection by recruiting neutrophils in a timely and coordinated matter {Rajarathnam, 2019 #753}. In order to avoid misunderstandings concerning IL-8's effects on cancer cells and neutrophils, we removed IL-8 addition experiments in figure 5. Thus in figure 5, we demonstrated that NETs could trap GC cells to form NETs-GC clusters, in which NETs facilitated GC cells extravasation and proliferation at metastatic sites. Still, thank you for your question.

11- IL-8 is used as a chemotactic factor in Fig.5, however IL-8 has been reported to promote NET formation in multiple papers (Brinkmann, 2004; Keshari, 2012; An, 2019; Nie, 2019 – to name only a few). It is therefore intriguing to see that the presence of IL-8 do no have any effect on the invasion of cancer cells cultured with non-IC neutrophils, considering that NETs should be formed

also in this condition (and this needs to be controlled). Another problem comes from Fig. 5B, where the basal invasion of GC cells is the same in all the condition. Indeed in Fig. 2A and 2D, there is a clear increase in the IC condition (Fig. 2A and 2D). The authors should explain this discrepancy.

Response Firstly, thank you for pointing out the careless mistake in figure 5. The discrepancy was mainly due to the difference in cell numbers we planted in transwell chamber among different experiments. In order to eliminate these ambiguities, we have carefully redone these experiments and the cell number had been elaborated in Methods part. Secondly, as mentioned in Major concern 10, we had removed IL-8 addition experiments in order to avoid misunderstandings concerning IL-8's effects on cancer cells and neutrophils.

12- In Fig. 5D, the authors use medium in the lower chamber for SEM. They need to explain why cells would be present in the medium and not on the lowest membrane of the Boyden chamber.

Response Thank you for your question. In figure 5, penetrated cells including GC cells and neutrophils could be observed both on the lowest membrane of the Boyden chamber and in the medium. The former was a typical experiment in which GC cells were stained with crystal violet and counted by a microscopy for invasive and migrative ability comparison. The latter was specially used for SEM to observe the interaction or trapping between GC cells and NETs because those mediums need to be plated on poly-L-lysine-coated coverslips and fixed in 2% glutaraldehyde for SEM observation. The detailed methods had been showed in Methods part.

13- Some cancer cells can promote the formation of NETs (Park, 2016; Lee, 2019). The ability of the GC cells used in this manuscript to promote NET formation needs to be assessed, because it could alter all the conclusion from the authors.

Response Thank you for raising concern. We have learnt the reference you have kindly provided and performed experiments to assess the ability of the GC cells to promote NETs formation. As a result, supplement figure 3G showed that little NETs were observed by immunofluorescence after coculturing MKN-45, MGC-803 and AGS with neutrophils, which was different from breast cancer reported by Park et al. {Park, 2016 #324} and Lee et al. {Lee, 2019 #86}. Still, thank you for your question and we had added those explanation to our discussion part with yellow highlight.

14- EMT is assessed in vivo in Fig. 3E and 3F thanks to different epithelial and mesenchymal markers. However, the pattern exposed in these representative pictures seems unlikely. For example, non-IC lungs should have no to little metastasis (from previous experiments), and indeed, from the histology of the lungs presented, the lungs look quite normal. However, all the cells in the lungs seems to be E-cad positive, which seems unlikely. Even more surprisingly, in the IC group, metastases should be present, and the histology of the lungs look quite normal as well, without any apparent metastasis, however all the lung cells are now E-cadherin negative. One can ask why all normal cells are E-cad positive in the non-IC group and why all normal cells are E-cad negative in the IC group. The same comments apply to the mesenchymal markers (N-cad, vimentin, snail). The staining pattern needs to be addressed and more importantly EMT needs to be assessed specifically in the cancer cells. This can be done by staining the cancer cells (they express luciferase, and luciferase antibodies can be used to detect them).

Response Thank you for kind suggestions. The problems you mentioned above could be led out by

two factors: 1) The resolution of IHC-P (immunohistochemistry- paraffin) we used before was too low that overstaining of these markers was prone to be magnified; 2) It is also our mistake to produce too much unspecific staining. To solve these problems, we performed immunofluorescence experiments for EMT markers (E-cadherin and N-cadherin) in GC cells coculturing with indicated neutrophils and in metastatic lesions from LM and PM nude mice that were injected by GC cells coculturing with indicated neutrophils (figure 3). Moreover, metastatic lesions resected from GC patients with LM or PM were collected for EMT markers examination by IHC (figure 3 and its result). In these results, HE was used to determine tumor cells (GC) and IHC-P or immunofluorescence were used to observe the alteration of EMT markers on tumor cells.

15- TGF-beta needs to be quantified in the serum of IC and non-IC patients to test its correlation with the presence of NETs. TGF-beta also needs to be quantified in IC and non-IC neutrophils culture.

Response Thank you for suggestions. We have quantified TGF- β 1 in the serum from control, Non-AIC and AIC patients by ELISA, and make a correlation analysis with the NETs level (figure 4 and its result). Also, TGF- β 1 was quantified in control, Non-AIC and AIC neutrophils supernatant (supplement figure 3 and its result). Moreover, we performed immunofluorescence co-staining of NETs and TGF- β 1 in neutrophils from control, Non-AIC, AIC and AIC + DNase I groups as well as metastatic lesions resected from GC patients with LM or PM (figure 4). As a result, the TGF- β 1 concentration was positively correlated with the level of NETs and TGF- β 1 was present with NETs release both *in vitro* and *in vivo* (figure 4E-G). While in the neutrophils supernatant, the TGF- β 1 concentration dramatically decreased as NETs levels did (supplement figure 3F, G), which thus had little effect on GC metastasis. By the way, NETs were hardly purified and subsequently little “free NETs” were detected in neutrophils supernatant in our study (supplement figure 3E, F). This result was something different from a number of studies concerning “free NETs”, in which free NETs not only could be detected, but also have been shown to bind to cancer cells {Monti, 2018 #745} {Monti, 2017 #746} {Najmeh, 2017 #743} {Park, 2016 #324}. This different result was probably because no NETs stimuli, such as calcium ionophore {Monti, 2018 #745} {Monti, 2017 #746}, LPS or PMA {Najmeh, 2017 #743} were used in our experiments. All neutrophils released NETs in our experiments were pathophysiological (AIC in patients or CP in nude mice) without any artificial stimuli addition. Still, thank you for your question and we had added those content to the results part with yellow highlight.

16- The authors show that neutrophils supernatant from IC neutrophils do not promote invasion and migration, and therefore conclude that direct contact between NET-forming neutrophils and GC cells is important. However, from the data presented, it is impossible to know if NETs are present in the supernatant (there should be some NETs as supernatant is commonly used by groups working on NET biology to assess the effect of “free NETs”). Moreover, NETs in the supernatant (“free NETs”) have been shown to bind to cancer cells and this need to be assessed and discussed (Najmeh, 2017; Monti, 2017 and 2018 - to name a few). Interestingly, it has also been previously published that NET-supernatant can promote breast cancer cells invasion *in vitro* (Park, 2016) and therefore, the differences observed need to be discussed.

Response Thank you for your suggestions. We had used SEM and ELISA (MPO-DNA level) to

examine whether there were any free NETs in the supernatant from AIC neutrophils. As a result, little “free NETs” were detected by SEM and ELISA and the level was quite low than that in serum and plasma (supplement figure 3 and its result). Also, the TGF- β 1 level in supernatant was also as low as “free NETs”(supplement figure 3 and its result). Therefore, we made a conclusion that “neutrophils supernatant from AIC patients did not promote GC invasion and migration”. While in those study concerning “free NETs binding to cancer cell” or “NETs-supernatant”, the isolated “free NETs or NETs-supernatant” were generated by calcium ionophore {Monti, 2018 #745} {Monti, 2017 #746}, LPS, PMA {Najmeh, 2017 #743} {Park, 2016 #324} or cancer cells {Park, 2016 #324} stimulation *in vitro*, which were very different from the methods in our study where NETs were detected without any stimulation except pathophysiological process such as AIC in patients or CP in nude mice. Those above differences had been discussed in our revised manuscript with yellow highlight texts. Thank you again

17- The authors show that IC-neutrophils supernatant does not promote invasion and migration of the cancer cells. It is important to test the presence of TGF-beta in the culture supernatants and its effect on EMT markers. Indeed, while all the experiments with TGF-beta inhibition are interesting, it requires more investigation. For example, is TGF-beta present on the NETs? Where is TGF-beta coming from?

Response Thank you for questions. As mentioned in question 15, the concentration of TGF- β 1 in AIC and Non-AIC neutrophils supernatant were quantified in supplement figure 3F and the level was as low as MPO-DNA level. We further investigated its effects on EMT markers, invasion and migration of GC cells in supplement figure 4. Also, figure 4E and F showed that TGF- β 1 positively correlated with the expression of NETs (MPO-DNA level) and was present with NETs release both *in vitro* and *in vivo* (figure 4G). Therefore, we assumed that TGF- β 1 was present with NETs release and mainly came from NETs in the context of infection and then in the NETs-GC clusters, the TGF- β signaling pathway of GC cells would be activated via direct contact with NETs. however, Due to the low concentration of NETs as well as TGF- β 1 in the supernatant, the AIC-neutrophils supernatant could not promote invasion and migration in our study.

18- The authors conclude from Figure 5 that after LPS treatment NETs can carry the cancer cells to infectious site through chemotaxis. There are mainly two problems on this conclusion. First, LPS is administered 5 days after cancer cell injection, and it is not clear whether at this point, CTCs are present or if the cancer cells extravasated in the lungs. Therefore, it is impossible to conclude whether NETs trap the cancer cells in the circulation to bring them to the metastatic site (and it is impossible to know where the NETs are formed, in the circulation or at the infectious site). Second, the authors do not assess properly chemotaxis of cancer cells at the infectious site considering that tail vein injection is used, which commonly leads to cancer cells being trapped in the lungs.

Response Thank you for your suggestions. To track GC cells that extravasated or present at the target organs, we labeled GC cells with GFP and injected them to establish LM and PM nude mice models. Then in the modeling of Sham, CP, CP + DNase I, we sacrificed those mice at post-modeling day 5 to 1) isolate neutrophils from peripheral blood and ascites fluid for NETs-GC clusters detection; 2) collect liver and peritoneum for GC extravasation or survival detection. As a result, NETs-GC clusters could be detected both in peripheral blood and ascites fluid in CP nude

mice whose neutrophils were stimulated to release NETs. Moreover, more GC cells extravasated and survived in the liver or peritoneum in CP nude mice (figure 5 and supplement figure 5). Therefore, our viewpoint was that NETs could trap the free GC cells in the peripheral blood or ascites fluid in the context of infection and help them extravasate and survive the metastatic site (liver or peritoneum). Also, all the lung metastasis model had been replaced by LM model to assess the metastatic ability of GC cells. Still thank you for your suggestion.

19- To test whether the NETs really trap the cancer cells to infectious site, the presence of CTCs needs to be quantified and a proper extravasation assay needs to be performed. Also, NETs have been shown to promote proliferation of low cycling cancer cells in the lungs (Albregues, 2018). The authors seem to indicate that NET do not alter proliferation of the cancer cells in vitro, but in vivo, the environment is different, and NETs could play a role. Therefore, it would be interesting to digest NETs with DNase I at later time points to assess NETs effect on proliferation at metastatic sites.

Response Thank you for raising concern. To solve these problems, we established LM and PM nude mice model with sham, CP or CP+DNase I to detect and compare GC cells in peripheral blood, ascites fluid and metastatic sites (figure 5 and supplement figure 5). Consequently, CP induced NETs trapped the GC cells in the peripheral blood or ascites fluid (NETs-GC clusters) and help GC cells extravasate and survive the metastatic site while less NET-GC clusters and GC cells were observed in sham or CP+DNase I groups. Then in supplement figure 6, Ki67 IHC assay showed that NETs enhanced proliferation of GC cells at metastatic lesions but could be counteracted by DNase I. Still, thank you for your question.

20- In this manuscript, IC is defined by a combination of clinical findings and results of examinations. Is it possible to be more specific and to score these criteria to highlight the threshold between IC and non-IC patients?

Response Thank you for your question. In our current study, postoperative complication was firstly classified by Clavien-Dindo classification {Clavien, 2009 #60} {Dindo, 2004 #61}, which has been adopted widely in clinical practice worldwide. Afterwards, a complications list proposed by Gastrectomy Complications Consensus Group {Baiocchi, 2019 #59} was used to define each complication, based on clinical complains, physical examination and radiology (X-ray, ultrasonography and computed tomography (CT)) reports. For example, duodenal or anastomotic leak was diagnosed by the presence of abdominal pain, fever (body temperature $\geq 38.0^{\circ}$), saliva or gastrointestinal contents in the drain or during relaparotomy, and radiologically by the contrast swallow test {Kim, 2015 #634}; Abdominal infection or abscess was diagnosed by the presence of abdominal pain, fever (body temperature $\geq 38.0^{\circ}$), pus in the drain or that abdominal fluid collection or abscess were found on CT or during relaparotomy {Climent, 2016 #788}. Those two above postoperative complications were defined as AIC. Pneumonia was defined as a newly developed infiltrates on the chest radiograph and positive results of bronchoalveolar lavage culture {Baba, 2016 #64} {Arozullah, 2001 #633} {Tu, 2017 #253}. Other complications including delayed gastric emptying or bowel obstruction requiring parenteral nutrition, pleural effusion requiring drainage, postoperative bleeding requiring both urgent transfusions and invasive treatment, myocardial infarction with patient's transfer to CCU/ICU/other critical care facility et al. were classified as Non-AIC.

Minor concerns:

1- Some sentences are not grammatically correct within the manuscript.

Response We apologized for these grammatical mistakes. In this revised manuscript, we had invited a professional English editor to help us polish this manuscript for correctness and readability. We hope the flow and language have been substantially improved.

2- Some acronyms need to be spell out properly (for example: R0 resection, TNFa, IL-8, CRP and PCT).

Response We are sorry for our carelessness. R0 resection represents (no cancer residue after resection microscopically. TNF- α is tumor necrosis factor α . CRP is C-reactive protein. PCT is procalcitonin. All the acronyms in our manuscript had been spell out full name at first time.

3- Within the introduction, the authors states that solid data exploring NETs-directed therapy for clinical practice is absent, which is not true. See the papers that the authors are citing (Cools-Lartigue, 2013; Tohme, 2016; Park, 2016; Albregues 2018; Lee, 2019)

Response Thank you for your correction. Cools-Lartigue et al. used DNase I or NEi treatment in animals with CLP-induced sepsis and found these treatment abrogated micrometastatic tumor formation{Cools-Lartigue, 2013 #311}. Tohme et al found DNase I or PAD4 targeting could attenuate surgical stress induced liver micrometastases by inhibiting the protumorigenic effects of NETs{Tohme, 2016 #312}. Park et al. found treatment with NET-digesting, DNase I-coated nanoparticles markedly reduced lung metastases in mice{Park, 2016 #324}. Albregues et al. showed inhibiting NET formation or digesting the NETs' DNA scaffold prevented conversion of single disseminated cancer cells to growing metastases in mouse models of breast and prostate cancer. Lee et al. found blockade of NET formation using a PAD4 pharmacologic inhibitor decreased omental colonization in mice. All these above studies concerning NETs-directed therapy were based on cells or animals experiments and offered a valuable reference for clinically exploring NETs-directed therapy. Thank you.

4- The authors collect peripheral blood samples from patients with or without postoperative ICs. It would be interesting to know which kind of ICs were involved (anastomotic leaks or pneumonia) and whether the isolated neutrophils are different, at least in their ability to form NETs.

Response In our revised manuscript, we focused on AIC that were consisted of gastrointestinal leak or abdominal abscess or infection, which was identified as an independent prognostic factor for RFS and OS. According to our current data, there was no significant difference in NETs release, MPO-DNA and TGF- β 1 levels among gastrointestinal leak or abdominal abscess or infection groups. Thank you.

5- Interestingly, elevated CRP and SAA were also associated with reduced disease-free survival for breast cancer patients. This paper should be cited as it seems to indicate a general mechanism in different type of solid cancer.

Response In our revised manuscript, we mainly focused on NETs-GC clusters induced metastasis as well as its potential mechanism. However, based on our current experimental data, it is hard to

generalize our proposed mechanism in other solid cancer. Elevated CRP and SAA were reported to associate with reduced disease-free survival for breast cancer patients {Winters-Stone, 2018 #756} {Takeuchi, 2017 #762}. But their underlying mechanism need to be further investigated.

6- In fig. 1E, the number of neutrophils is quantified, but it is not written how. Did the authors used FACs and Ly6G staining, cells morphology, something else?

Response: Thank you for your question. In our study, the number of neutrophils from ascites were quantified by SYSMEX XNL-350 in the department of Laboratory Medicine, Renji Hospital. One of our research members, Enhao Zhao, finished these experiments in this laboratory. Specifically, anticoagulant tubes were used to collecting ascites fluid by drainage tube, and the test was completed within 4 hours after collection in strict accordance with the SYSMEX XNL-350 instructions.

7- In fig. 1G, NETs are quantified, but it not written how. Considering that a major problem of the manuscript is the detection of NETs, this need to be clearly stated.

Response: Thank you for your question. With regard to NETs quantification in immunofluorescence, we mainly referenced Albregues et al.'s study {Albregues, 2018 #85}, in which the NETs were counted as extracellular citrullinated H3 (Cit-H3) positive cells at least three representative immunofluorescence images (from 2 neighboring sections) per sample. Then neutrophils were counted as MPO or Ly6G positive cells at least three representative immunofluorescence images (from 2 neighboring sections) per sample. The percentage of NET-forming neutrophils was calculated using the formula: (number of NET-forming neutrophils/number of neutrophils)*100. For NETs-GC clusters quantification in SEM, it was determined by the mean percentage of the neutrophils (white arrow) releasing webs (green arrow) binding with GC cells (red arrow) in at least ten representative views.

8- The authors mention using a TGF-β1 agonist which never seems to be used.

Response TGF-β1 was used in supplement figure 4 to explore its effects on GC cells EMT, invasion and migration.

9- TGF-beta have been shown to modulate neutrophils phenotype (Fridlender, 2009) toward a “N1” or “N2” phenotype. The authors should assess the effect of TGF-beta of the ability to induce or inhibit NETs in IC and non-IC neutrophils.

Response Thank you for your suggestion. We performed these coculturing experiments such as coculturing neutrophils with GC cells (supplement figure 3) or TGF-β1 (data not show). However, neither of them could stimulate neutrophils to release NETs

10- A clinical score of sepsis is quantified in Fig 6B and supp 6C, but the method for such a quantification do not appear in the manuscript.

Response Thank you for your question. The method for mice sepsis score quantification referred to Manley et al. 's study {Manley, 2005 #340}. Specifically, the maximum possible score of six comprised the presence of the following signs: lethargy, piloerection, tremors, periorbital exudates, respiratory distress, and diarrhea. Each symptom was noted as one point for its presence and as 0 point for its absence. We are sorry for our carelessness and these specific descriptions had been

added to our Methods part.

11- NETs digestion leads to sepsis in the CP model indicating that NETs are critical in eliminating pathogens. However, neutrophils can also phagocyte and degranulate to counteract an infection, and it might be important to analyze their ability to do so in the context of a DNase I treatment and TGF-beta inhibition.

Response In our study, we mainly focused on NETs-GC clusters induced metastasis and the way to suppress metastasis while not concurrently aggravated sepsis. Then in our CP modeling with LM and PM metastasis *in vivo* experiments, we observed that both DNase I or LY 2157299 (TGF- β signaling inhibitor) administration significantly reduced metastatic burden (figure 6C, D, G, H), while NETs digestion leads to severe sepsis (figure 6B), no matter how phagocytosing or degranulating function in neutrophils. In our opinion, those experiments involving the other anti-infection functions of neutrophils would not change our conclusion that meticulous operation to decrease postoperative AIC or targeting downstream effectors of NETs such as TGF- β signaling in the context of NETs-GC clusters could be used to prevent metastasis in locally advanced GC patients. Thank you.

12- The authors show that NETs can promote wound healing of cancer cells *in vitro*. However, it has been previously shown that NETs can counteract wound healing in patients with diabetes (Ling Wong, 2015). This need to be discussed.

Response The discrepancy in wound healing mediated by NETs is intriguing for investigation. In my opinion, the reasons probably are: 1) the physiological environment in cancer differ from that in diabetes, which is defined as a group of metabolic diseases clinically characterized by hyperglycemia that results from defects in insulin action, insulin secretion, or both; 2) cancer cells undergo distinguished biological behaviors compared to normal cells. However, these hypotheses need confirmation through well-designed experiments. In order to avoid ambiguity on our proposed mechanism that NETs-GC cluster facilitate GC metastasis, we removed wound healing experiments and retain tranwell experiments in our revised manuscript. Thank you.

13- IL-8, TNF-a and CRP detection were performed in hospital's routine laboratory. The authors need to be more specific and provide protocol and antibody catalog numbers.

Response Thank you for your suggestion. In our revised manuscript, we only test IL-8, a potent chemotactic factor {Van Damme, 1988 #771} {Walz, 1987 #772} in ascites fluid from GC patients. This ascites fluid IL-8 was detected by ELISA (MultiSciences, 70-EK108HS-96) according to the manufacturer's protocol in which the amounts of IL-8 were measured spectrophotometrically by the absorbance at 450 nm.

14- The concentration of DNase I used is not written for *in vitro* experiment.

Response We are sorry for our carelessness. The concentration of DNase I used for *in vitro* experiments was 0.25 mg/ml (Roche, 11284932001).

15- For migration and invasion assay, the methods indicate a 8 μ m porous membrane, while Supp. Fig. 4 indicate a 4 μ m porous membrane.

Response We are sorry for our carelessness and the porous membrane was 8 μ m. Thank you for

your prudent review.

16- When the authors are quantifying migration/invasion of the cancer cells, are they using morphological differences to distinguish between invading cancer cells and invading neutrophils? Response In our transwell assay, we did distinguish cancer cells from neutrophils morphologically. In our preliminary neutrophils transwell assay, neutrophils (1×10^6 cells) were loaded on the upper chamber of the transwell without GC cells for invasion and migration. We found penetrated neutrophils was morphologically different from cancer cells after crystal violet staining. Neutrophils were more round with less antenna compared to cancer cells. Therefore, in our transwell assay for GC cells, we counted those stained cells in the flat and irregular morphology with more antenna. Those detailed description concerning transwell assay had been added to Methods part in our revised manuscript.

Reviewer #2 (Remarks to the Author):

The manuscript "Neutrophil extracellular traps produced during postoperative infection facilitate recurrence of gastric cancer" investigated the mechanisms of postoperative infectious complications promote gastric cancer recurrence and decrease long-term survival. There were several comments on this manuscript:

1. the most important are the survival of infectious complication affect the patients' survival. Patients with complications usually with more advanced cancer, so the comparison of patients with or without complication need balance the clinicopathologic data first, especially for stages.

Response Thank you for your suggestion. In order to control confounding factors such as pStage, we performed multi-analysis to identify the independent risk factors for OS and RFS. As a result, supplement figure 1A showed the results of Cox proportional hazards regression model to identify independent prognostic factors for OS and RFS. In this multi-analysis, GC patients with complication (Clavien-Dindo \geq II) was identified as an independent prognostic risk factor for both OS and RFS. Their corresponding survival curves grouped by complication group (C group) and no complication group (NC group) were presented graphically in supplement figure 1B. Notably, the curves stratified by pStage (I, II, III) showed that the significantly shorter 5-year OS and RFS due to complication only existed in II and III cohort (supplement figure 1C). Furthermore, stratified analysis indicated that those significant negative effects on prognosis mainly resulted from postoperative AIC in those cohorts (supplement figure 1D). Still, thank you for your advisement.

2. The figures were so small and difficult to see the figures well.

Response We are sorry for those low-resolution figures. In the revised manuscript, we had uploaded high-resolution figures with 300dpi for review. Thank you.

3. There was not too much innovations.

Response In this article, we demonstrated that in the context of infection, NETs could trap GC cells to form NETs-GC clusters, in which NETs facilitate GC cells metastasis. The hypothesis concerning infection, NETs and metastasis helps to better understand postoperative AIC inducing poor prognosis and highlight the importance to decreasing postoperative AIC after gastrectomy,

especially in locally advanced GC patients. Moreover, we provide a therapeutically potential target to mitigate the adverse oncologic consequences in GC patients undergoing postoperative AIC and not aggravate sepsis concurrently.

Reviewer #3 (Remarks to the Author):

In the manuscript of Xia and colleagues, the authors seem to present a consistent story using a variety of intro and preclinical in vivo model to draw a functional link between infectious complication-associated neutrophil NET formation and dissemination of gastric cancer cell lines. Such observations would be consistent that infectious complication associated with gastric cancer resection decreased survival.

The reason why my above statement is highly hypothetical results from the insurmountable challenge for this reviewer to deal with highly inappropriate small sizing of the data figures combined with their low resolution, which the authors chose for their submission. This prevents this assessor from evaluating the authors' interpret of their data presented and hence to assess the extent by which the authors' texts appropriately reflect the data shown. In addition, this reviewer is also not in a position to assess the quality of the data presented in the first instance.

Notwithstanding my major critique above, as a minor comment, this author criticizes the authors for not having included scale bars in their photomicrographs.

Response We are sorry for those low-resolution figures. In the revised manuscript, we had uploaded high-resolution figures with 300dpi for review. Plus, the scale bars had also been added in the figures. Thank you for your further review.

Reviewers' Comments:

Reviewer #1:

Remarks to the Author:

I would like to thank the authors for answering clearly to all my comments. A tremendous effort has been done to ameliorate this manuscript up to "nature communications" standard.

However, I still have some minor comments:

In fig. S3A, it seems that co-culture of cancer cells with neutrophils and/or cancer cells with neutrophils supernatant was used. This is still unclear from the legend, the cartoon and the material and methods section. In this figure, proliferation of cancer cells co-cultured with neutrophils is assessed using a CCK-8 assay. This assay assess proliferation of both cancer cells and neutrophils in the dish as it was done on the co-culture. Therefore, the author cannot conclude on the effect of neutrophils on cancer cell proliferation specifically.

It is also unclear what is the difference between the assays used in S3B and 5A.

Some sentences in the manuscript are still hard to understand. Words are missing and some sentences are grammatically incorrect. For example:

"In order to confirm above findings in vivo, liver metastasis (LM) and peritoneal metastasis (PM) nude mice models as described in Methods were introduced." There is no verb in this sentence.

"As a result, coculturing with NETs-formation neutrophils from AIC groups promoted metastasis of GC cells, and the treatments that specifically inhibiting NETs formation could significantly decrease metastasis". I think here the authors refer to NET-forming neutrophils

In Fig. 3D, the staining for MPO and Cit-H3 is strange. Indeed, it seems that there are a lot of Cit-h3 staining for only a few MPO staining, indicating that most of the cells expressing cit-h3 are not neutrophils.

The authors conclude "Altogether, these results revealed that NETs trapping free GC cells to form NETs-GC clusters facilitated them extravasation and survival at metastatic sites". While I agree that their results suggest this, it was not shown in this manuscript. The only way to visualize that trapped cancer cells extravasate is to use intravital imaging. I am not asking the authors to use intravital imaging, but to tone down their conclusion.

It is unclear from the presented data if the authors are looking at extravasation at day 5, as we do not know when the first cancer cells stopped in the vasculature of the metastatic organ. Therefore, it is possible that at this time point, the authors also analyze the effect of cancer cell proliferation.

The authors show that "NETs promoted proliferation of GC cells in metastatic sites and this effect would be lost when NETs digested by DNase I or TGF- β signaling inhibited by LY 2157299" From their in vivo data, it seems that TGFbeta signaling is also having an effect on cancer cell proliferation. However, the authors show that NETs do not have any effect on cancer cell proliferation in vitro. These suggest that TGFbeta signaling, independently of NETs promote proliferation in vivo? Then why the NET-associated TGFbeta is not having any effect on proliferation?

Also, if NET-associated TGFbeta promotes proliferation (which is still unclear when comparing in vitro and in vivo results), maybe the effect from NETs on invasion/migration is just the results from more cancer cells in the transwell assay.

The authors write "inhibition of PAD4 in Pad4^{-/-} transgenic" in the discussion, which to my knowledge has not been done, because it should not be relevant.

Reviewer #2:

Remarks to the Author:

The manuscript by Xia et al. used clinical samples and different experimental in vitro and in vivo to propose that NETs, produced during infectious complications after gastrectomy for gastric cancer. Although there were several studies showed that the effect of neutrophils and NETs in infection

and inflammation-induced metastasis, the authors used the in vivo models which made this research more interesting.

I saw the authors have revised the manuscript according to the reviewers comments. I think it have been good enough to published on Nature Communications.

Reviewer #4:

Remarks to the Author:

In this manuscript, Xia et al explored the cellular mechanisms of NETs-GC clusters in gastric cancer metastasis using animal model system which induces abdominal sepsis and display similar features of postoperative abdominal infectious complication (AIC) in human patients.

I am reviewing a revised version of the manuscript originally submitted to NCOMM. All-reviewers have focused their comments on the data quality and novelty. More importantly, Reviewer 1 concerned that animal models used in this manuscript lacked details or controls and were miss-interpreted. Although the authors improved the quality of their animal studies in the resubmitted manuscript, I agree with these concerns and also feel it is still a major issue with this paper.

The main point of this study claims to demonstrate that NETS-formation neutrophils can trap free GC cells to form NETs-GC clusters and promote the GC metastasis. However, the major shortcoming is that there are not any convincing data which show the direct interaction between NETs-Neutrophils and injected gastric cancer cells in vivo. In their conclusion and model figure (Figure 7), it was stated that the NETs from neutrophils are released at the infectious sites and abdominal cavity as well as in peripheral blood. Authors focused to show the NET formation and the presence of Neutrophil-NETs at those sites. There is one piece of data examining the injected gastric cancer cells in the metastatic lesions (Figure 5G & 5H), however, this immunostaining data in different sections cannot corroborate the authors' claim that the Neutrophil-NETs can trap the cancer cells at all. Cancer cells communicate indirectly through stimulated signals such as chemokines and/or cytokines from immune cells and the roles of Neutrophil-NET formation during cancer metastasis have been previously reported in other types of cancers as authors also mentioned. Therefore, providing the direct evidence of the presence of Neutrophil-NET and gastric cancer cell clusters after the sepsis induction and further investigation of the clusters in GC metastasis is extremely critical in vivo especially after the CP or CLP operations.

Immunofluorescence images are too small and the quality is too poor, although those data are very important to reach to a conclusion that authors claims. Especially in the IF staining in tissues, it is hard to assess whether signals of NETs positive for the Cit-H3 are real and the TGF beta1 signals are truly present in neutrophils. In Figure 4G, authors stained for TGF beta1 and Cit-H3 in peripheral blood and ascites fluid and in clinical tissues. The fluorescent signals of TGF beta and Cit-H3 are not present in same cells meaning that it is not clear whether the TGF beta1 secreted from NET-presenting Neutrophils counteracts the NETs-GC complex formation and even the TGF beta1 is secreted from the NET-presenting Neutrophils. Although the TGF beta1 level in serum or ascites may be correlated with the level of MPO-DNA, it does still not mean the TGF beta1 is secreted from the NET-presenting Neutrophils because the Neutrophils are not the only one source of TGF beta1 secretion. The authors performed an in vivo TGF beta inhibition study using LY2157299 which is a TGF beta receptor 1 inhibitor, supposedly controlling the receptor activity in cancer cells rather than the actual TGF beta secretion. However again... It is not clear whether the results from this animal study can support the roles of TGF beta1 secreted from the Net-presenting Neutrophils.

In conclusion, I feel the resubmitted manuscript does not provide conclusive evidence to support the functions of AIC-induced Neutrophil NETs-GC complex in GC metastasis that authors claim. Supporting this interpretation/conclusion would require robust orthogonal experiments.

REVIEWER COMMENTS

Reviewer #1 (Remarks to the Author):

I would like to thank the authors for answering clearly to all my comments. A tremendous effort has been done to ameliorate this manuscript up to “nature communications” standard.

Response Thank you for your inspiring and encouraging comments. All our research colleagues are grateful to your questions and suggestions, that give our study a great of substantial amelioration. Thanks!

However, I still have some minor comments:

In fig. S3A, it seems that co-culture of cancer cells with neutrophils and/or cancer cells with neutrophils supernatant was used. This is still unclear from the legend, the cartoon and the material and methods section. In this figure, proliferation of cancer cells co-cultured with neutrophils is assessed using a CCK-8 assay. This assay assess proliferation of both cancer cells and neutrophils in the dish as it was done on the co-culture. Therefore, the author cannot conclude on the effect of neutrophils on cancer cell proliferation specifically.

Response First of all, we are sorry for our indistinct description on cell viability assay. The result that NETs did not alter the proliferation of gastric cancer (GC) cells *in vitro* remained to be disputable, as you pointed that we assessed proliferation of both GC cells and neutrophils in the dish. Then in this revised manuscript, the cell viability assay was corrected as “GC cells were seeded in 96-well plates at a density of 5000 cells per well. After incubation for 4 hours (GC cells adherence), the medium was replaced by 100 μ L conditioned medium such as PBS, indicated neutrophils (1×10^5), DNase I (0.25 mg/ml, Roche, 11284932001) or neutrophils supernatant and then the medium was changed every 8 hours. At indicated times thereafter, culture mediums were removed and then after gently washing with PBS three times, 10% CCK-8 (Dojindo, C0038) with RPMI 1640 were added to incubate for a further 1 hour. Then cell viability was determined by CCK-8 at a wave length of 450 nm. Three independent experiments were carried out.”. Due to few suspended neutrophils adherence after PBS washing, GC cells proliferation could be assessed specifically by CCK-8. As a result, NETs-formation neutrophils promoted GC cells proliferation, while neutrophils without NETs formation or NETs digested by DNase I failed (supplement figure 3A, B). Again, thanks for your correction. We had revised our “cell viability assay” methods and added those proliferation assay results with yellow highlighted words in this revised manuscript.

It is also unclear what is the difference between the assays used is S3B and 5A.

Response We are sorry for our indistinct description. In previous supplement figure 3B, after GC cells seeding 6 hours for adherence, indicated neutrophils or control treatment were added to upper wells for 8 hours incubation. Here, we merely focused

on the motility of GC cells after indicated neutrophils or control treatment. While in figure 5A, a mixture of GC cells and indicated neutrophils were seeded on the upper wells simultaneously, in which NETs could trap GC cells to form NETs-GC clusters for subsequent motility. More notably, the penetrated NETs-GC clusters could be observed directly in the lower chamber by scan electron microscopy (SEM), that had been showed in figure 5C. Here, we focused on the motility of NETs-GC clusters. In order to avoid these misunderstanding or confusion, we had removed those ambiguous schematic supplement figure 3B in this revised manuscript, and added more detailed and clear description into “Transwell assay” methods part to clarify the difference with yellow highlighted words.

Some sentences in the manuscript are still hard to understand. Words are missing and some sentences are grammatically incorrect. For example:

“In order to confirm above findings in vivo, liver metastasis (LM) and peritoneal metastasis(PM) nude mice models as described in Methods were introduced.” There is no verb in this sentence.

“As a result, coculturing with NETs-formation neutrophils from AIC groups promoted metastasis of GC cells, and the treatments that specifically inhibiting NETs formation could significantly decrease metastasis”. I think here the authors refer to NETforming neutrophils

Response Thanks. Another professional English editor was invited to improve the readability and fluency of our manuscript. However, for second sentence, the treatments including DNase I, NE inhibitor and PAD4 inhibitor were used to specially inhibit NETs rather than neutrophils. Therefore, we corrected this sentence as “As a result, coculturing with NETs-formation neutrophils from AIC groups promoted GC cells metastasis, and the treatments that specifically inhibiting NETs could significantly alleviate those metastasis”. Still, thank you for your careful review.

In Fig. 3D, the staining for MPO and Cit-H3 is strange. Indeed, it seems that there are a lot of Cit-h3 staining for only a few MPO staining, indicating that most of the cells expressing cit-h3 are not neutrophils.

Response Thank you. This problem probably resulted from the low-resolution and color aberration of Fig. 3D. In this revised manuscript, we submitted high-resolution representative pictures for Fig. 3D.

The authors conclude “Altogether, these results revealed that NETs trapping free GC cells to form NETs-GC clusters facilitated them extravasation and survival at metastatic sites”. While I agree that their results suggest this, it was not shown in this manuscript. The only way to visualize that trapped cancer cells extravasate is to use intravital imaging. I am not asking the authors to use intravital imaging, but to tone down their conclusion.

Response Thank you for your suggestion and understanding. As you stated, using intravital imaging to visualize that trapped GC cells extravasate would provide more convincing evidence. But regrettably, we could hardly provide those “visualize” data

in our current article due to lack of intravital imaging equipment. Nevertheless, we referred to Sheri A. C. McDowell⁴, Juwon Park⁷ and Asaf Spiegel⁸'s studies concerning tumor cells extravasation experiments and performed those experiments in this revised manuscript. Specifically, we injected GFP-MKN-45 cells into spleen (liver metastasis nude mice model) and abdominal cavity (peritoneal metastasis nude mice model) in sham, CP (25G) and CP (25G) + DNase I nude mice respectively. Twenty-four hours (post-modeling day 1) after GC cell injection, mice were euthanized and then the liver and peritoneum samples were harvested, processed for H&E and immunofluorescence staining to observe GC cell extravasation and implantation in these distant organs. As a result, more GC cells extravasated into the liver and implanted in the peritoneum in CP (25G) group compared to that in the other two groups at post-modeling day 1 (figure 5E, F). This was also corresponding to the existing results in figure 5G and H (post-modeling day 5), which further indicated that more extravasated and implanted GC cells could proliferate in liver and peritoneum of CP (25G) group, as you stated in next comment. Therefore, our conclusion is "Altogether, these above results implicated that AIC-induced NETs could trap free GC cells to form NETs-GC clusters and then contributed to GC cells extravasation, implantation and proliferation in those metastatic organs." (had toned down and yellow highlighted).

Still, it is regretful that direct intravital evidence concerning NETs-GC clusters extravasation were not available in this manuscript. But through analyzing liver and peritoneum samples among sham, CP (25G) and CP (25G) + DNase I groups at post-modeling day 1, day 5 and day 20, we successively observed that, in the aid of infection induced NETs, more GC cells extravasated, implanted and proliferated in these distant organs (liver and peritoneum) for metastatic foci formation, which provided substantial data for our hypothesis and conclusion. Thank you for suggestion.

It is unclear from the presented data if the authors are looking at extravasation at day 5, as we do not know when the first cancer cells stopped in the vasculature of the metastatic organ. Therefore, it is possible that at this time point, the authors also analyze the effect of cancer cell proliferation.

Response Thank you. These data from samples of post-modeling day 5 might be not enough to examine GC cells extravasation. So in this revised manuscript, we referred to Sheri A. C. McDowell⁴, Juwon Park⁷ and Asaf Spiegel⁸'s studies concerning tumor cells extravasation experiments and added those extravasation experimental data as follows. As a result, through examining GFP signaling in liver and peritoneum samples at post-modeling day 1, we found more GC cells extravasated into the liver and implanted in the peritoneum in CP (25G) group as compared to that in the other two groups (figure 5E, F). This was also corresponding to the results in figure 5G and 5H (post-modeling day 5), which further indicated that more extravasated and implanted GC cells proliferated in the liver and peritoneum via CP (25G) induced NETs formation. Furthermore, through analyzing liver and peritoneum samples at post-modeling day 1, day 5 and day 20, we successively observed that, in the aid of NETs, more GC cells extravasated, implanted and proliferated in the distant organs

(liver and peritoneum), all of which were very important events in the metastatic program. Still, thanks for your helpful question.

The authors show that “NETs promoted proliferation of GC cells in metastatic sites and this effect would be lost when NETs digested by DNase I or TGF- β signaling inhibited by LY 2157299” From their *in vivo* data, it seems that TGF β signaling is also having an effect on cancer cell proliferation. However, the authors show that NETs do not have any effect on cancer cell proliferation *in vitro*. These suggest that TGF β signaling, independently of NETs promote proliferation *in vivo*? Then why the NET-associated TGF β is not having any effect on proliferation?

Response First of all, we apologized for our confused description and incorrect protocol concerning cell viability assay again, which brought out those inconsistent proliferation results between *in vitro* and *in vivo*. As responded to your first minor comment, we used corrected methods to assess GC cells proliferation in this revised manuscript. As a result, NETs-formation neutrophils promoted GC cells proliferation *in vitro*, while neutrophils without NETs formation or NETs digested by DNase I failed, which were consistent with the proliferation assay results *in vivo*. These above data had been added to this revised manuscript with yellow highlight. Again, thanks for your correction.

Also, if NET-associated TGF β promotes proliferation (which is still unclear when comparing *in vitro* and *in vivo* results), maybe the effect from NETs on invasion/migration is just the results from more cancer cells in the transwell assay.

Response It is a good question! Firstly, NETs did promote proliferation *in vitro* after we used corrected “cell viability assay” according to your suggestion. Thus, it was reasonable to suspect that the effect of NETs on proliferation would interfere invasion/migration results in the transwell assay. However, in our transwell assay, we only take 8 hours incubation time to observe those invasion/migration results. While in our cell viability assay result, there were no significant difference among those indicated neutrophils or inhibitor treatment during the first 24 hours (supplement figure 3A, B). Those results revealed that proliferation mediated by NETs at the first 24 hours would not significantly affect the outcomes of the invasion/migration assay. Therefore, we deduced that the effect from NETs on invasion/migration is not the results from proliferated cancer cells in the transwell assay.

The authors write “inhibition of PAD4 in Pad4^{-/-} transgenic” in the discussion, which to my knowledge has not been done, because it should not be relevant.

Response Sorry. This is my fault. We had deleted this error.

Finally, our research team do appreciate your suggestion and comments!

Reviewer #2 (Remarks to the Author):

The manuscript by Xia et al. used clinical samples and different experimental *in vitro*

and in vivo to propose that NETs, produced during infectious complications after gastrectomy for gastric cancer.

Although there were several studies showed that the effect of neutrophils and NETs in infection and inflammation-induced metastasis, the authors used the in vivo models which made this research more interesting.

I saw the authors have revised the manuscript according to the reviewers comments. I think it have been good enough to published on Nature Communications.

Response Thank you for your positive comments.

Reviewer #4 (Remarks to the Author):

In this manuscript, Xia et al explored the cellular mechanisms of NETs-GC clusters in gastric cancer metastasis using animal model system which induces abdominal sepsis and display similar features of postoperative abdominal infectious complication (AIC) in human patients.

I am reviewing a revised version of the manuscript originally submitted to NCOMM. All-reviewers have focused their comments on the data quality and novelty. More importantly, Reviewer 1 concerned that animal models used in this manuscript lacked details or controls and were miss-interpreted. Although the authors improved the quality of their animal studies in the resubmitted manuscript, I agree with these concerns and also feel it is still a major issue with this paper.

Response Thank you for your comments. Our research team will do our best to ameliorate our data or provide reasonable and convincing explanation to those concerns. Still, we appreciate your review and criticisms.

The main point of this study claims to demonstrate that NETS-formation neutrophils can trap free GC cells to form NETs-GC clusters and promote the GC metastasis. However, the major shortcoming is that there are not any convincing data which show the direct interaction between NETs-Neutrophils and injected gastric cancer cells in vivo. In their conclusion and model figure (Figure 7), it was stated that the NETs from neutrophils are released at the infectious sites and abdominal cavity as well as in peripheral blood. Authors focused to show the NET formation and the presence of Neutrophil-NETs at those sites. There is one piece of data examining the injected gastric cancer cells in the metastatic lesions (Figure 5G & 5H), however, this immunostaining data in different sections cannot corroborate the authors' claim that the Neutrophil-NETs can trap the cancer cells at all. Cancer cells communicate indirectly through stimulated signals such as chemokines and/or cytokines from immune cells and the roles of Neutrophil-NET formation during cancer metastasis have been previously reported in other types of cancers as authors also mentioned. Therefore, providing the direct evidence of the presence of Neutrophil-NET and gastric cancer cell clusters after the sepsis induction and further investigation of the clusters in GC metastasis is extremely critical in vivo especially after the CP or CLP operations.

Response Thank you for this worthwhile question. As Reviewer #1 have addressed, intravital imaging could provide convincing evidence to demonstrate NETs-Neutrophils interacting with injected GC cells. But regrettably, the intravital imaging equipment was not easily available or borrowed under current strict epidemic control policy. However, figure 5A, B, C demonstrated the NETs-GC clusters migrated from upper wells to lower chamber *in vitro* (a mixture of GC cells and indicated neutrophils were added to upper wells for NETs-GC clusters motility assay). Notably, figure 5C exhibited the representative images of NETs-GC clusters via SEM. Supplement figure 5H and 5I exhibited that neutrophils isolated from CP (25G) nude mice could form NETs-GC clusters in the specific transwell assay. Moreover, figure 5E, F, G, H were successive section of samples to corroborate that NETs-forming neutrophils facilitated GC cells extravasation, implantation and proliferation at metastatic sites. In consideration of that several studies had corroborated NETs could trap or attach tumor cells *in vivo* and *in vitro*¹⁻³ and neutrophils could directly contact circulating tumor cells (CTCs) of breast cancer to form CTC-neutrophil clusters *in vivo*⁹, we concluded that AIC-induced NETs could trap free GC cells to form NETs-GC clusters and then contributed to GC cells extravasation, implantation and proliferation in those metastatic organs (had toned down according to Reviewer #1's suggestion). Overall, our study mainly focused on investigating the oncologic consequence after GC cells trapped by NETs and clinically exploring a feasible and harmless therapeutic strategy (TGF- β inhibition) to reduce metastasis in those GC patients undergoing postoperative infection, as simply neutrophils depletion or NETs inhibition in a mouse model of polymicrobial sepsis would result in higher rates of systemic microbial dissemination⁵.

Still, our research team appreciated your suggestion on this viewpoint.

Immunofluorescence images are too small and the quality is too poor, although those data are very important to reach to a conclusion that authors claims. Especially in the IF staining in tissues, it is hard to assess whether signals of NETs positive for the Cit-H3 are real and the TGF beta1 signals are truly present in neutrophils. In Figure 4G, authors stained for TGF beta1 and Cit-H3 in peripheral blood and ascites fluid and in clinical tissues. The fluorescent signals of TGF beta and Cit-H3 are not present in same cells meaning that it is not clear whether the TGF beta1 secreted from NET-presenting Neutrophils counteracts the NETs-GC complex formation and even the TGF beta1 is secreted from the NET-presenting Neutrophils. Although the TGF beta1 level in serum or ascites may be correlated with the level of MPO-DNA, it does still not mean the TGF beta1 is secreted from the NET-presenting Neutrophils because the Neutrophils are not the only one source of TGF beta1 secretion.

Response Thank you for your criticism concerning our figure quality. Here, we replaced those low-quality figures with more representative high-quality figures for those IF staining in tissues, such as figure 3D and 4H. Then for figure 4G, although TGF- β 1 signaling was not completely overlap (a majority of TGF- β 1 overlap) with Cit-H3, it was much stronger in NETs-presenting neutrophils than those without NETs, such as control, Non-AIC or AIC+DNase I groups. In view of TGF- β 1 level in serum

and ascites fluid positively correlating with the level of MPO-DNA, we supposed that TGF- β 1 level was also positively correlated with NETs release. Actually, it has been widely established that NETs containing DNA absorbs several proteins, such as MPO, MMP9 or neutrophil elastase⁶. Likewise, TGF- β 1 probably was one of proteins that attached to DNA webs by absorption or NETs-presenting neutrophils secretion (figure 4G).

All in all, we agreed with your viewpoint that neutrophils were not the only source of TGF- β 1 or TGF- β 1 might not be secreted only from the NETs-presenting neutrophils. Nevertheless, our main point was that TGF- β 1 level was positively correlated with NETs release and AIC induced NETs would promote GC metastasis via NETs-GC clusters, in which TGF- β signaling of GC cells could be activated by NETs-attached TGF- β 1.

The authors performed an *in vivo* TGF beta inhibition study using LY2157299 which is a TGF beta receptor 1 inhibitor, supposedly controlling the receptor activity in cancer cells rather than the actual TGF beta secretion. However again... It is not clear whether the results from this animal study can support the roles of TGF beta1 secreted from the Net-presenting Neutrophils.

Response Thank you for this important question again. As we mentioned above, we did not conclude that TGF- β 1 was secreted only from the NETs-presenting neutrophils. Our main point was TGF- β 1 level was positively correlated with NETs release and TGF- β signaling of GC cells would be significantly activated dependent on NETs-GC clusters. Moreover, by performing an *in vivo* TGF- β inhibition study using LY 2157299, we just expected to explore a promising strategy to reduce NETs-promoted metastasis while not to undermine NETs dependent bacterial clearance. Still, elucidating TGF- β 1 origin or release mechanism are indeed an interesting and valuable subject in our future research work.

In conclusion, I feel the resubmitted manuscript does not provide conclusive evidence to support the functions of AIC-induced Neutrophil NETs-GC complex in GC metastasis that authors claim. Supporting this interpretation/conclusion would require robust orthogonal experiments.

Response We still appreciate your criticisms and comments that greatly improve the quality of our manuscript. Although intravital imaging experiments and TGF- β 1 origin or release mechanism were not completely achievable in our current study, our results did provide convincing evidence that NETs-formation neutrophils promoted GC metastasis and TGF- β inhibition could significantly reduce these metastasis while not undermining the anti-infection of NETs. Our conclusion proposed a prospect to mitigate the adverse oncologic consequences as well as not increase polymicrobial sepsis in GC patients undergoing postoperative AIC.

Reference

- 1 Albrengues J, Shields MA, Ng D, Park CG, Ambrico A, Poindexter ME *et al.*

Neutrophil extracellular traps produced during inflammation awaken dormant cancer cells in mice. *Science* 2018; 361.

2 Cools-Lartigue J, Spicer J, McDonald B, Gowing S, Chow S, Giannias B *et al.* Neutrophil extracellular traps sequester circulating tumor cells and promote metastasis. *J Clin Invest* 2013.

3 Lee W, Ko SY, Mohamed MS, Kenny HA, Lengyel E, Naora H. Neutrophils facilitate ovarian cancer premetastatic niche formation in the omentum. *J Exp Med* 2019; 216: 176-194.

4 McDowell SAC, Luo RBE, Arabzadeh A, Doré S, Bennett NC, Breton V *et al.* Neutrophil oxidative stress mediates obesity-associated vascular dysfunction and metastatic transmigration. *Nature Cancer* 2021; 2: 545-562.

5 Meng W, Paunel-Gorgulu A, Flohe S, Hoffmann A, Witte I, MacKenzie C *et al.* Depletion of neutrophil extracellular traps in vivo results in hypersusceptibility to polymicrobial sepsis in mice. *Crit Care* 2012; 16: R137.

6 Papayannopoulos V, Metzler KD, Hakkim A, Zychlinsky A. Neutrophil elastase and myeloperoxidase regulate the formation of neutrophil extracellular traps. *J Cell Biol* 2010; 191: 677-691.

- 7 Park J, Wysocki RW, Amoozgar Z, Maiorino L, Fein MR, Jorns J *et al.* Cancer cells induce metastasis-supporting neutrophil extracellular DNA traps. *Sci Transl Med* 2016; 8: 361ra138.
- 8 Spiegel A, Brooks MW, Houshyar S, Reinhardt F, Ardolino M, Fessler E *et al.* Neutrophils Suppress Intraluminal NK Cell-Mediated Tumor Cell Clearance and Enhance Extravasation of Disseminated Carcinoma Cells. *Cancer Discov* 2016; 6: 630-649.
- 9 Szczerba BM, Castro-Giner F, Vetter M, Krol I, Gkoutela S, Landin J *et al.* Neutrophils escort circulating tumour cells to enable cell cycle progression. *Nature* 2019; 566: 553-557.

Reviewers' Comments:

Reviewer #1:

Remarks to the Author:

I would like to thank the authors for their work and answering clearly my comments (the highlighted yellow text made it easy to review their work).

While the authors have answered all my comments, I still have some minor comments that might need to be addressed to avoid wrong conclusions.

1) The authors claim "Due to few suspended neutrophils adherence after PBS washing, GC cells proliferation could be assessed specifically by CCK-8."

While I agree that PBS washes might remove neutrophils from the plate, this needs to be assessed properly as a control. Indeed, it is possible that in some conditions neutrophils are more adherent than in other conditions, and that could impact result from the experiment (especially when the authors focus on cancer cells and neutrophils clusters). The authors use GFP expressing cancer cells, that can be used to quantify the number of cancer cells specifically for example. Other methods are also possible.

2) The authors claim "altogether, these above results implicated that AIC-induced NETs could trap free GC cells to form NETs-GC clusters and then contributed to GC cells extravasation, implantation and proliferation in those metastatic organs"

This conclusion needs to be tone down furthermore if no additional experiments are used. Indeed, the authors do not show properly the trapping of cancer cells with NETs in vivo, as the only way to assess a proper "trapping" is live imaging in vivo. Again, I do not expect the authors to use intravital imaging, but they need to tone down this conclusion, using "these above results suggested...".

Another important point is that the authors do not assess proliferation in vivo at these time points (1 and 5 days after cancer cell injection). Therefore, it is possible that the increase of cancer cell extravasation is solely responsible for the higher number of cancer cells detected at day 5. If the author wants to claim that proliferation is involved in vivo at day 5, they need to assess it properly, or they need to tone down their conclusion at this point. Indeed, the authors have looked at cell proliferation in vivo, only later in their manuscript, thanks to Ki67 staining in Fig. S6E (which is not the same time point than in Fig. 5-H)

3) The legends on Fig. 5F and 5H are not readable due to their size.

4) Comments on the authors response to reviewer 4.

I agree with reviewer 4 that the source of TGFbeta is still obscure. The authors answered that NET-DNA "absorbs" proteins such as MPO, MMP9 and NE, which is true, but normal neutrophils are also positive for these proteins. Here it seems that normal neutrophils are not positive for TGFbeta.

In Fig 4G, while there is the presence of TGFbeta within NETs in the AIC group, TGFbeta seems to disappear after DNase I treatment. If TGFbeta was produced by AIC neutrophils, one would assume that some TGFbeta would be detected within the cells after DNase I treatment. The data presented by the authors are not strong enough to claim that neutrophils secrete TGFbeta within NETs.

Beside these minor comments and if reviewer 4 is convinced, I believe that this manuscript can be published in Nature Communications and that the scientific community will be interested by the author's findings.

Reviewer #4:

Remarks to the Author:

This reviewer agrees with the authors' concern regarding the technical difficulties. The methodology is not strong enough to provide definitive evidence of the direct interaction between neutrophils and GC cells. Nevertheless, authors have improved the data quality in other supporting data.

Reviewer #1 (Remarks to the Author):

I would like to thank the authors for their work and answering clearly my comments (the highlighted yellow text made it easy to review their work).

Response Thank you very much. Your queries and comments do give our study a great of improvement. We are all grateful to your prudent reviews.

While the authors have answered all my comments, I still have some minor comments that might need to be addressed to avoid wrong conclusions.

1) The authors claim “Due to few suspended neutrophils adherence after PBS washing, GC cells proliferation could be assessed specifically by CCK-8.”

While I agree that PBS washes might remove neutrophils from the plate, this needs to be assessed properly as a control. Indeed, it is possible that in some conditions neutrophils are more adherent than in other conditions, and that could impact result from the experiment (especially when the authors focus on cancer cells and neutrophils clusters). The authors use GFP expressing cancer cells, that can be used to quantify the number of cancer cells specifically for example. Other methods are also possible.

Response Thank you for your rigorous comments and reminding. We performed GFP expressing gastric cancer cells proliferation assays and assessed proliferation by fluorescence microscope according to your advice. As was expected, after coculturing GFP expressing MKN-45 and MGC-803 cells with indicated neutrophils or control, significantly more GFP gastric cancer cells were observed in AIC group compared to control and Non-AIC group (figure 1 for the Reviewers only). It indicated that neutrophils from AIC group promoted gastric cancer proliferation while those from control and Non-AIC group did not. Therefore, the conclusion concerning proliferation assay is reliable. Those fluorescence figures have been uploaded as Figure 1 for the Reviewers only. Thank you.

Here is the method: GFP expressing GC cells were seeded in 96-well plates at a density of 5000 cells per well. After incubation for 4 hours (GC cells adherence), the medium was replaced by 100 μ L conditioned medium such as PBS, indicated neutrophils (1×10^5), DNase I (0.25 mg/ml, Roche, 11284932001) or neutrophils supernatant and then the medium was changed every 8 hours. At indicated times thereafter, culture mediums were removed and then gently washed with PBS three times. Finally, cell viability was assessed by fluorescence microscope observation. Three independent experiments were carried

figure 1 for the Reviewers only: Representative immunofluorescence of coculturing GFP expressing MKN-45 (upper panel) and MGC-803 cells (down panel) with indicated neutrophils or control (n= 3 biologically independent experiments).

2) The authors claim “altogether, these above results implicated that AIC-induced NETs could trap free GC cells to form NETs-GC clusters and then contributed to GC cells extravasation, implantation and proliferation in those metastatic organs”

This conclusion needs to be tone down furthermore if no additional experiments are used. Indeed, the authors do not show properly the trapping of cancer cells with NETs in vivo, as the only way to assess a proper “trapping” is live imaging in vivo. Again, I do not expect the authors to use intravital imaging, but they need to tone down this conclusion, using “these above results suggested...”.

Response Still, we appreciated your considerate suggestion and understanding. We had further toned down our conclusion that “these above results suggested that AIC-induced NETs could trap free GC cells to form NETs-GC clusters and then contributed to GC cells extravasation and implantation in those metastatic organs” with yellow highlighted words in this revised manuscript. Thank you again.

Another important point is that the authors do not assess proliferation in vivo at these time points (1 and 5 days after cancer cell injection). Therefore, it is possible that the increase of cancer cell extravasation is solely responsible for the higher number of cancer cells detected at day 5. If the author wants to claim that proliferation is involved in vivo at day 5, they need to assess it properly, or they need to tone down their conclusion at this point. Indeed, the authors have looked at cell proliferation in vivo, only later in their manuscript, thanks to Ki67 staining in Fig. S6E (which is not the same time point than in Fig. 5-H)

Response Thank you for your correction. We had deleted those description concerning proliferation in this part.

3) The legends on Fig. 5F and 5H are not readable due to their size.

Response We apologized for the small legends on Fig. 5F and 5H. The larger legends for easy readability on Fig. 5F and 5H had been resubmitted in this revised supplement figure 6A and 6B.

4) Comments on the authors response to reviewer 4.

I agree with reviewer 4 that the source of TGFbeta is still obscure. The authors answered that NET-DNA “absorbs” proteins such as MPO, MMP9 and NE, which is true, but normal neutrophils are also positive for these proteins. Here it seems that normal neutrophils are not positive for TGFbeta.

In Fig 4G, while there is the presence of TGFbeta within NETs in the AIC group, TGFbeta seems to disappear after DNase I treatment. If TGFbeta was produced by AIC neutrophils, one would assume that some TGFbeta would be detected within the cells after DNase I treatment. The data presented by the authors are not strong enough to claim that neutrophils secrete TGFbeta within NETs.

Response We apologized for the poor quality of Fig 4G. Actually, TGFbeta was a bit positive in the control group and after DNase I treatment (figures showed below). In the control group, the magenta for TGFbeta overlapped green (DNA) and red (Cit-H3). In the AIC+DNase I group, we adjusted the tone of magenta (TGFbeta) and then TGFbeta could be visible within the cells after DNase I treatment. The revised Fig 4G had been submitted. Thank you.

G

TGF- β 1

TGF- β 1

figure 2 for the Reviewers only: Representative immunofluorescence staining images of TGF- β 1 in the neutrophils isolated from peripheral blood (upper panel) and ascites fluid (down panel) of control and AIC+DNase I groups;

Beside these minor comments and if reviewer 4 is convinced, I believe that this manuscript can be published in Nature Communications and that the scientific community will be interested by the author's findings.

Response Thanks a lot!

Reviewer #4 (Remarks to the Author):

This reviewer agrees with the authors' concern regarding the technical difficulties. The methodology is not strong enough to provide definitive evidence of the direct interaction between neutrophils and GC cells. Nevertheless, authors have improved the data quality in other supporting data.

Response Thanks a lot!